



# GARRLiC and LIRIC: strengths and limitations for the characterization of dust and marine particles along with their mixtures

**Alexandra Tsekeri[1], Anton Lopatin[2], Vassilis Amiridis[1], Eleni Marinou[1,3], Julia Igloffstein[4], Nikolaos Siomos[3], Stavros Solomos[1], Panagiotis Kokkalis[1], Ronny Engelmann[4], Holger Baars[4], Myrto Gratsea[5], Panagiotis I. Raptis[5,6], Ioannis Binietoglou[7], Nikolaos Mihalopoulos[5,8], Nikolaos Kalivitis[1,8], Giorgos Kouvarakis[8], Nikolaos Bartsotas[9], George Kallos[9], Sara Basart[10], Dirk Schuettemeyer[11], Ulla Wandinger[4], Albert Ansmann[4], Anatoli P. Chaikovsky[12], and Oleg Dubovik[2]**

[1]{Institute for Astronomy, Astrophysics, Space Applications and Remote Sensing, National Observatory of Athens, Athens, Greece}

[2]{Laboratoire d' Optique Atmosphérique, Universite de Lille, Lille, France}

[3]{Laboratory of atmospheric physics, Physics Department, Aristotle University of Thessaloniki, Greece}

[4]{Leibniz Institute for Tropospheric Research, Leipzig, Germany}

[5]{IERSD, National Observatory of Athens, Athens, Greece}

[6]{Physikalisch-Meteorologisches Observatorium Davos/World Radiation Center (PMOD/WRC), Davos Dorf, Switzerland}

[7]{National Institute of R&D for Optoelectronics, Magurele, Ilfov, Romania}

[8]{Environmental Chemical Processes Laboratory, University of Crete, Heraklion, Greece}

[9]{University of Athens, School of Physics, Athens, Greece}

[10]{Barcelona Supercomputing Center, Barcelona, Spain}

[11]{European Space Agency}

[12]{Institute of Physics, NAS of Belarus, Minsk, Belarus}

Correspondence to: Alexandra Tsekeri (atsekeri@noa.gr)





## 1 Abstract

The **G**eneralized **A**erosol **R**etrieval from **R**adiometer and **Li**dar **C**ombined data algorithm (GARRLiC) and the **LI**dar-**R**adiometer **I**nversion **C**ode (LIRIC) provide the opportunity to study the aerosol vertical distribution by combining ground-based lidar and sun-photometric measurements. Here, we utilize the capabilities of both algorithms for the characterization of Saharan dust and marine particles, along with their mixtures, in the South-Eastern Mediterranean during the "**CHAR**acterization of **A**erosol mixtures of **D**ust and **M**arine origin **Exp**eriment (CHARADMExp)". Three case studies are presented, focusing on dust-dominated, marine-dominated and dust/marine mixing conditions. GARRLiC and LIRIC achieve a satisfactory characterization for the first case in terms of particle microphysical properties and concentration profiles. The marine-dominated and the mixture cases are more challenging for both algorithms, although GARRLiC manages to provide more detailed microphysical retrievals compared to AERONET, while LIRIC effectively discriminates dust and marine in its concentration profile retrievals.

## 1    Introduction

The importance of studying the vertical distribution of aerosol plumes is prominent in regional and climate studies, since it can effectively change the radiative properties of the atmosphere and the presence of clouds (e.g. Pérez et al., 2006a; Solomon et al., 2007). Ground-based monitoring of the aerosol vertical structure is effectively performed with the synergy of passive and active remote sensing instruments, in particular with multi-wavelength sun-photometers and lidars. The sun-photometer provides the columnar properties of the particles (e.g. Dubovik and King, 2000a; Dubovik et al., 2006), whereas the lidar is capable of providing vertical profiles of the backscatter and extinction coefficients, along with vertical profiles of the particle microphysical properties, mainly for the fine mode (e.g. Müller et al., 2015). The combination of active with passive remote sensing has been tried so far mainly by using the sun-photometer measured aerosol optical depth (AOD) as ancillary information for the lidar retrieval (e.g. Fernald et al., 1972; Ansmann et al., 2011; 2012). GARRLiC (Lopatin et al., 2013) and LIRIC (Chaikovsky et al., 2016) algorithms go a step further and use deeper synergies: the LIRIC approach derives the particle concentration profiles from the lidar measurements, considering the columnar microphysical properties derived separately from the sun-photometer; GARRLiC



advances the method even more, combining for the first time both sun-photometer and lidar
measurements for the retrieval of the particle microphysical properties. As discussed in detail
in Lopatin et al. (2013), combining the sun-photometer intensity measurements with the
backscatter lidar information seems to result in better sensitivity to the particle shape, as well
as the ability to retrieve the refractive indices of fine and coarse particles separately, along with
extracting the vertical distribution of the fine and coarse particle concentrations. Moreover, it
can potentially provide higher accuracy for cases of low aerosol loadings, compared with the
intensity-only retrieval.
GARRLiC and LIRIC have been developed in the framework of the Aerosols, Clouds and Trace
gases Research Infrastructure (ACTRIS, http://www.actris.eu/), utilizing the capabilities of
combined European stations of the AErosol RObotic NETwork (AERONET, Holben et al.,
1998) and the European Aerosol Research Lidar Network (EARLINET, Pappalardo et al.,
2014). Both algorithms have been tested for a variety of aerosol types and their mixtures. For
example, LIRIC has been tested for dust and volcanic aerosols (Wagner et al., 2013),
dust/pollution mixture (Tsekeri et al., 2013), dust, pollution and mixture of dust/smoke and
pollution (Granados-Muñoz et al., 2014; 2015; Papayannis et al., 2014), and smoke/pollution
mixture (Kokkalis et al., 2016). LIRIC has also been used to study dust transport events and
dust modeling performance over Europe (Binietoglou et al. 2015, Granados-Munoz, 2016), as
well as to evaluate air quality models (Siomos, et al. 2017).  GARRLiC has been tested for dust
and smoke (Lopatin et al., 2013) and dust aerosols (Bovchaliuk et al., 2016).
GARRLiC and LIRIC input and output data is shown in Fig. 1, while short descriptions are
given herein: LIRIC algorithm uses the particle microphysical properties provided in the
AERONET product as a-priori information in the inversion of the lidar measurements for
retrieving the aerosol volume concentration profiles. Using lidar measurements of elastic
backscatter at three wavelengths of 355, 532, and 1064 nm, LIRIC retrieves the volume
concentration profiles of fine and coarse particles, whereas considering also the cross-polarized
lidar signal at 532 nm the coarse mode can be disentangled into its spherical and non-spherical
components. The error estimation of the retrieved profiles is provided as well. Both LIRIC and
GARRLiC retrievals assume that key aerosol properties vary smoothly (e.g. aerosol
concentration varies smoothly with height), but otherwise do not constrain the absolute values
of the retrieved quantities. In this way the algorithms exclude solutions that are mathematically
possible, but contain unrealistic oscillations in the retrieved properties (see also Dubovik, 2004;



Dubovik and King, 2000). GARRLiC algorithm combines the sun-photometer sun and sky
measurements at four wavelengths (at 440, 670, 870 and 1020 nm) and up to 35 scattering
angles, with the vertically-resolved lidar measurements of the elastic backscatter at three
wavelengths (at 355, 532, and 1064 nm). The algorithm calculates the size distribution,
spherical particle fraction and spectral complex refractive index, separately for fine and coarse
particles, considering them constant along the atmospheric column, and the volume
concentration profiles of fine and coarse particles. The retrieval uncertainties of the
microphysical parameters are provided as well, and the profile retrieval uncertainties are
currently under development. The volume concentration below the lowest height of the lidar
signals is considered to be constant. Moreover, in case of a dominant mode (e.g. for pure dust
cases), the algorithm is set to retrieve the aerosol characteristics for one mode only. GARRLiC
and its updates are available for download at http://www.grasp-open.com/doc/ch04.php#grasp-
manager, as part of the GRASP code (Dubovik et al., 2014).
In case of multi-mode aerosol mixtures an inherent deficiency of both algorithms is the number
of aerosol modes retrieved, with LIRIC considering three modes (fine particles, coarse spherical
and coarse non-spherical particles) and GARRLiC considering two modes (fine and coarse
particles). We need to highlight here that LIRIC retrieves three modes only for the volume
concentration profiles, whereas otherwise it uses the AERONET products, providing for
example a common spectral refractive index for all modes (Fig. 1). Both algorithms work well
for individual aerosol components or mixtures of (mainly) fine (e.g. pollution) and (mainly)
coarse (e.g. dust) particles, but they should not be able to fully characterize the mixture
components in case of more than one fine or coarse mode in the mixture, as in smoke/pollution
or dust/marine mixture cases. For the latter, LIRIC should provide an effective characterization
for the volume concentration profiles, since it derives the coarse spherical (hydrated marine)
particles and the non-spherical (dust) particles, but the characterization is not expected to be
satisfactory for the particle microphysical properties.
In our study, we apply GARRLiC and LIRIC for cases of dust, marine and dust/marine mixture
during the CHARADMExp campaign in the South-Eastern Mediterranean. This is the first time
a detailed characterization of marine and marine mixtures with dust along the atmospheric
column is performed for the area. So far, various studies have tried to characterize the aerosol
radiative properties in the Mediterranean with satellite or ground-based AOD measurements
(e.g. di Sarra et al., 2008; Kazadzis et al., 2009; Papadimas et al., 2012). Unfortunately, they





fail to overcome their limitations such as the non-realistic assumptions for the aerosol
absorption properties and the lack of information of the real vertical aerosol structure (Mishra
et al., 2014). The kind of characterization presented here is important for application in future
satellite missions not only for the Mediterranean, but for large parts of the globe where dust and
marine particles are present, as in the Atlantic Ocean (e.g. Prospero, 1996).
The CHARADMExp campaign and the three cases (i.e. mainly dust, marine/pollution mixture
and dust/marine/pollution mixture) are presented in section 2. The methodology followed in
our work is presented in section 3, the GARRLiC and LIRIC results are shown in section 4 and
finally our conclusions are given in section 5.

## 11    2    Overview of the CHARADMExp campaign and datasets

CHARADMExp was an experimental campaign of ESA, implemented by the National
Observatory of Athens (NOA), aiming at the characterization of dust and marine particles along
with their mixtures (http://charadmexp.gr). The campaign took place at the ACTRIS Finokalia
station (35.338°N, 25.670°E) on the north coast of Crete, in Greece (Fig. 2), from 20 June to
20 July 2014. The station is situated at the top of a hilly elevation (252 m above sea level) and
it is a background site with no human activities occurring at a distance shorter than 15 km,
making the station ideal for monitoring natural aerosols mainly of desert and marine origin. The
area is characterized by the existence of two well-distinguished seasons equally distributed
throughout the year: the dry season from April to September and the wet season from October
to April, with the first one characterized mainly by winds of N/NW direction (Central and
Eastern Europe and Balkans) carrying smoke and long range transported anthropogenic
pollution to the area (Sciare et al., 2008; Vrekoussis et al., 2005), and the second one with less
pronounced N/NW winds and important transport from the Sahara desert (S/SW winds;
occurrence up to 20%). Dust transport is less frequent during the dry period, however
commonly observed (e.g. Papadimas et al., 2005), characterized by a transportation pattern
through the free troposphere and weaker vertical mixing of the dust layers (e.g. Kalivitis et al.,

28    2007).



## 2.1 Instruments and methods

### 2.1.1 Lidar

The Polly[XT] OCEANET lidar (Engelmann et al., 2016) operated at a 24/7 basis during CHARADMExp, measuring aerosol loads in the boundary layer and the free troposphere. The system was provided by the Leibniz Institute for Tropospheric Research (TROPOS - http://www.tropos.de). It employs 3 backscatter channels (at 355, 532 and 1064 nm), 2 Raman extinction channels (at 387 and at 607 nm), 2 depolarization channels (at 355 and 532 nm) and one water–vapor channel (at 407 nm). The lidar is housed in a container and can be operated under various climatic conditions. The full description of the original lidar system can be found in Althausen et al. (2009) and in Engelmann et al. (2016). More about the network of Polly systems (i.e., Polly[NET]) can be found in Baars et al. (2016).

The information close to the surface is very important for our study, especially for the marine particle characterization, since the marine particles reside mostly below 1 km (e.g. Ho et al., 2015). Unfortunately, this is also the lidar "overlap region", with large uncertainty for the lidar backscattered signal due to its partial collection from the telescope (e.g. Wandinger and Ansmann, 2002). Polly[XT] OCEANET far-field (FF) signal full overlap is at ~800 m (Engelmann et al., 2016) and it operates two near–field (NF) channels utilizing a separate 50–mm refractor telescope at a distance of 120 mm from the axis of the laser beam, providing a full overlap at 150 m above surface at 532 and 607 nm. The NF measurements are not used in the input of GARRLiC and LIRIC, since both algorithms require the complete set of wavelengths provided by the lidar during CHARADMExp only for the FF measurements. Nevertheless, we use the NF measurements to perform overlap correction in the FF signals, as described in Engelmann et al. (2016), and this allowed us to use the FF-corrected lidar signals from ~550 m, instead of 800 m. In future efforts we plan to utilize the additional information provided by our new Polly[XT] lidar system currently installed at Finokalia station, measuring NF signals at both 355 and 532 nm, by performing the signal gluing technique for NF and FF signals at 355 and 532 nm and overlap correction for the FF signal at 1064 nm.

### 2.1.2 Sunphotometer

The CIMEL CE318 sunphotometer is the instrument used in the AERONET sunphotometer network, with more than 250 units worldwide. The technical specifications of the instrument are given in detail by Holben et al. (1998). Taking into account all the information about the




instrument precision and calibration precision (Holben et al., 1998) the accuracy of the AOD
measurements is estimated to be of the order of $\pm0.02$ in the UV and $\pm0.01$ in the Visible
regarding the level 2 (cloud-screened and quality-assured) data. In the current analysis we
utilized the level 1.5 products (i.e., automatically cloud cleared but may not have final
calibration applied) for the LIRIC retrieval, since the level 2 data were not available in the time
ranges selected for the retrievals. For the GARRLiC retrieval we used the sun and sky multi-
angle measurements at four wavelengths (440, 670, 870 and 1020 nm) (Dubovik and King,

8    2000).

### 9    2.1.3   Surface in situ

The GARRLiC retrieved size distribution is evaluated against the surface measurements of the
Scanning Mobility Particle Spectrometer (SMPS). SMPS provides the fine particle number size
distribution at ~9 - 848 nm (nominal) radius. Unfortunately, there were no size distribution
measurements for the coarse particles at Finokalia station during CHARADMExp. Note that
for a direct comparison of SMPS number size distribution (in $cm^{-3}$) with the GARRLiC
volume size distribution retrievals (in $\mu m^3 \mu m^{-2}$) we first have to calculate the SMPS volume
size distribution (in $\mu m^3 cm^{-3}$) and then to multiply it with the extent of fine particles in the
column, derived by the collocated lidar measurements.
Moreover, we evaluate the particle concentration derived from GARRLiC and LIRIC at the
surface level with the surface in situ measurements of the particular matter for particles with
diameters less than 10 $\mu m$ ($PM_{10}$). The $PM_{10}$ is continuously measured at Finokalia station with
an Eberline FH 62 I-R (Eberline Instruments GmbH) particulate monitor (Gerasopoulos et al.,
2006). GARRLiC and LIRIC retrieve the particle concentration for a wider size range (up to 15
$\mu m$ in radius, or 30 $\mu m$ in diameter), thus their $PM_{10}$ values are calculated using the respective
volume percentages for particles with radius less than 5 $\mu m$.
In order to compare the in situ measured size distribution and mass concentration with
GARRLiC and LIRIC ambient retrievals, we need to take into account the particle drying
applied to surface measurements. The in situ instruments dry the sampled air by adiabatic
compression during the sampling through their inlets and by the radiant heat from the lights
inside the instruments. The size and mass of the ambient particles thus changes, especially in
case of hygroscopic particles in humid conditions (e.g. Snider and Petters, 2008). For the size
distribution we evaluate this effect qualitatively (see section 4.2 and 4.3). For the $PM_{10}$





comparison we calculate the "dry" GARRLiC and LIRIC $PM_{10}$, using the particle hygroscopic
growth (i.e., the ratio of the ambient to dry particle size, $f_g$) as shown in Eq. 1:

$$PM_{10_d} = f_g^{-3} PM_{10_a} \qquad (1)$$

where $d$ and $a$ denote the dry and ambient particles, respectively.
We derive $f_g$ for different relative humidity (RH) values using the hygroscopicity parameter $\kappa$
(Petters and Kreidenweis, 2007) as shown in Eq. 2:

$$f_g = \left(1 + \kappa \frac{RH}{100-RH}\right)^{\frac{1}{3}} \qquad (2)$$

For the cases analysed herein, we consider a $\kappa$ value of 0.4 to be characteristic for particles in
the south-eastern Aegean Sea (Bezantakos et al., 2013). A more detailed treatment of comparing
dry in situ measurements with ambient remote sensing retrievals is out of the scope of this
analysis, but it is very important when combining these different techniques (e.g. Tsekeri et al.,

10  2017).

**2.2   Models**
**2.2.1   Source-receptor analysis**
The origin of the examined aerosol layers at the Finokalia station is investigated with the use
of source-receptor computations derived with dispersion modelling tools. The corresponding
emission sensitivity (i.e. the residence time of the tracer particles inside the lowest tropospheric
layers) is calculated from backward Lagrangian simulations with the atmospheric dispersion
model FLEXPART-WRF (Brioude et al., 2013). The dispersion model is offline coupled with
the WRF_ARW atmospheric model (Skamarock et al., 2008). The spatial resolution of WRF is
12×12 km and we use its hourly outputs to drive the FLEXPART runs. This configuration
allows the simulation of meso-γ scale circulations that play an important role for the planetary
boundary layer properties and for the regional and local scale transport of the particles. The
backward FLEXPART runs are performed for 5-day periods and we assume a release of 40000
tracer particles from each arriving layer over the Finokalia station. The modelled retroplume
maps show the spatial distribution of the tracer particle residence time below 1 km. Thus, the
areas showing longer residence times in these maps indicate the source areas/origin of the
particles arriving at the specific heights above Finokalia station.





### 2.2.2  Desert dust model

Desert dust emissions and transport are described with the BSC-DREAM8b model (Nickovic et al., 2012; Pérez et al., 2006a; Basart et al., 2012a). The BSC-DREAM8b model is embedded into the Eta/NCEP atmospheric model and solves the mass balance equation for dust, taking into account the different processes of the dust cycle (i.e., dust emission, transport and deposition). The updated version of the model includes a source function based on the 1 km USGS land use data, 8 particle size bins (0.1–10 μm radius range), and dust-radiative feedbacks.The present analysis utilize the BSC-DREAM8b dust simulations for the period from 20 June to 20 July 2014 with hourly output. The initial state of dust concentration in the model is defined by the 24 h forecast from the previous day model run. The NCEP Final Analyses (at $1^o \times 1^o$ horizontal resolution) at 00:00 UTC are used as initial conditions and boundary conditions at intervals of 6 h. Moreover, the model configuration includes 24 Eta vertical layers extending up to approximately 15 km in the vertical. The resolution is set to 1/3° in the horizontal.

### 2.2.3  Sea-salt model

Sea salt emissions and transport are described with the atmospheric model RAMS-ICLAMS (Solomos et al., 2011). The model is an enhanced version of RAMS (Pielke et al., 1992; Cotton et al., 2003) and it includes a full description of the sea salt lifecycle in the atmosphere. The parameterization of sea salt emission is based on the white-cap formation for the entrainment of sea salt spray in the atmosphere (Monahan et al., 1986), taking also into account the effects of RH on the size distribution of the particles (Zhang et al. 2005).  Sea salt flux close to the coastline is also calculated in the model following the parameterizations of Leeuw et al. (2000) and Gong et al. (2002). The dry and wet removal processes are treated with the corresponding schemes described in Seinfeld and Pandis (1998). The simulated sea salt mass is represented with a bimodal lognormal distribution. The first (accumulated) mode has a mean diameter of 0.36 μm and a geometric dispersion of 1.80.  The second (coarse) mode has a mean diameter of 2.85 μm and the geometric dispersion is 1.90.

### 3  Results

In order to demonstrate the GARRLiC and LIRIC capabilities in characterizing events with dust and marine particles, we analyse in detail three cases acquired during CHARADMExp at



Finokalia. The first case is a relatively moderate dust episode with low amount of marine and
continental particles, the second is a low-AOD marine and continental plume and the last is a
mixture of dust, marine and continental particles. Source-receptor simulations are used to derive
the particle origin and characterize the air masses. Then, we compare the optical properties
retrieved from GARRLiC and LIRIC, as well as the collocated Klett retrievals (Klett, 1985).
The GARRLiC and LIRIC/AERONET fine mode size distributions and $PM_{10}$ concentrations
are compared with surface in situ measurements. Finally, the dust and marine concentration
profiles are compared with the corresponding profiles from BSC-DREAM8b and RAMS-
ICLAMS models.
**3.1  Dust-dominated case**
On June 26 the Polly$^{XT}$ measurements of volume depolarization ratio at 532 nm showed the
advection of non-spherical particles (volume depolarization ratio at 532 nm of 0.15-0.2), at
height ranges extending from close to the ground up to 5-6 km (Fig. 3a) and an AOD at 440 nm
of 0.4. Model simulations also support our observations: dust transport simulations using the
BSC DREAM8b model indicate Saharan dust transport to Finokalia. As shown by the
FLEXPART footprints in Fig. 3b, the particles reaching from the ground up to 2 km have
possible near-surface sources at the West Sahara region, with potential mixing of marine and
continental particles from the western Mediterranean region, the Balkans and Greece, while the
particles arriving at 3-6 km are most likely dust from the Sahara desert between 0°-10° E and
25°-35° N. The presence of dust particles is indicated from AERONET as well, with Ångstrom
exponent at 440/870 nm of ~0.1, sphericity parameter <2.3 % and a coarse-mode dominated
size distribution. These values are characteristic for dust particles, as reported in the 8-year
global AERONET climatology of Dubovik et al. (2002).
Considering that the atmospheric column is dominated by dust (as shown in the coarse mode
dominated AERONET size distribution), we performed the one-mode GARRLiC inversion.
For both GARRLiC and LIRIC we used the lidar measurements at 4-6 UTC (red box in Fig.
3a) and the sun-photometer measurements at 4:54 UTC. Our results show that GARRLiC and
LIRIC backscatter and extinction coefficient profiles at 355, 532 and 1064 nm agree quite well
within the LIRIC uncertainties with small differences seen below ~550 m, in the lidar
incomplete overlap region (first and second row in Fig. 4). Figure 4 shows also the comparison
of GARRLiC backscatter and extinction coefficients with the ones produced with the Klett



method (Klett, 1985). The Klett profiles are cut above 5 km, since the low signal to noise ratio
of the day-time lidar measurements introduces large uncertainty to the Klett retrievals above
that height. For the Klett retrievals we used an extinction-to-backscatter ratio, or "lidar ratio"
(LR) of 40 sr for 532 and 1064 nm and of 47 sr for 355 nm, which result in extinction coefficient
profiles that closely reproduce the sun-photometer-measured AODs at 340, 500 and 1020 nm
(i.e. 0.42, 0.42 and 0.38), respectively. The uncertainty in the assumed lidar ratios are taken into
account by considering a 20 % uncertainty in the backscatter retrievals (Fig. 4, third row). The
agreement of GARRLiC with Klett retrievals is considered satisfactory, with differences for the
backscatter coefficient to be within the Klett retrieval uncertainty, and for the extinction
coefficient to be less than 30% at heights above 550 m. Figure 4 shows also the NF retrievals
at 532 nm, providing information of the particle properties down to 150 m: In particular, we
see a decrease in the particle backscatter and extinction coefficients near the surface, which is
not retrieved by GARRLiC or LIRIC due to missing NF information as discussed in section

14   3.1.1.

A special feature seen in GARRLiC, LIRIC and Klett backscatter profiles is the larger
backscatter at 532 than 355 nm. This is not usual for dust particles, but it has been reported
before: Veselovskii et al. (2016) have shown a similar spectral dependence for dust during the
study of SaHAran Dust Over West Africa (SHADOW) campaign, which they attributed to large
dust particle spectral variation of the imaginary part of the refractive index. More specifically,
they managed to reproduce this backscatter spectral dependence with imaginary part values of
0.005-0.05 at 355 nm and 0.005 at 532 nm. Although these values are not the same with the
retrieved 0.001 at 355 nm and 0.0005 at 532 nm for our case (Fig. 5 –bottom, right), the
backscatter spectral dependence can be a combination of the effect that different factors have
on the backscattered light, as the size, shape or orientation of the dust particles.
Figure 5 shows good agreement between GARRLiC and AERONET retrievals (the latter used
in the LIRIC retrieval), within the GARRLiC retrieval uncertainties. Differences are seen only
for the real part of the refractive index, which for GARRLiC is at ~1.45, at the low end of the
dust climatological value range of 1.48±0.05-1.56±0.03 as reported in Dubovik et al. (2002).
This value though is much lower than expected for dust from West Sahara in situ measurements,
reporting values of 1.55-1.65 (e.g. Kandler et al., 2007), and it may be due to the marine particle
mixture at lower heights, with real part of refractive index of ~1.35. An important feature of the
GARRLiC retrieval is the spectral dependence of the single scattering albedo (SSA), showing



the characteristic increase of dust absorption in the ultraviolet (Fig. 5, up right) (Otto et al.,
2007). Moreover, the GARRLiC size distribution agrees well with surface in-situ SMPS
measurements for the fine mode, showing a very small volume concentration for fine particles.
The SMPS number size distribution is converted to $\mu m^3 \mu m^{-2}$ for a direct comparison with the
GARRLiC and AERONET product, as described in section 2.1.3: For this conversion we
consider that mainly the first 2 km contain fine particles due to the mixing of marine and
continental particles with dust there (Fig. 3b). Moreover, due to the low RH at the surface of
16%, we do not expect differences between the GARRLiC ambient size distribution and the
SMPS dry measurements.
The concentration profiles from GARRLiC and LIRIC are in excellent agreement at heights >1
km (Fig. 6a). LIRIC retrieves fine and coarse mode profiles, whereas GARRLiC considers only
one mode, dominated by coarse particles. The LIRIC coarse mode is comprised only of non-
spherical particles. Figure 6b (left) shows the comparison of GARRLiC and LIRIC dust particle
profiles with the BSC DREAM8b model. For this comparison we consider all particles in
GARRLiC and LIRIC profiles to be dust particles. Furthermore, we multiply them with the dust
density of $2.6 \, g \, cm^{-3}$ (Reid et al., 2003) to convert the volume concentration ratio (in ppb) to
dust mass concentration (in $\mu g \, m^{-3}$). Although the shapes agree well, the BSC DREAM8b
model values are lower than GARRLiC and LIRIC, by a factor of 2. The BSC DREAM8b
underestimation when comparing to LIRIC is consistent with the findings of Binietoglou et al.
(2015) for relative low dust concentrations (as is the case here). The underestimation is shown
in the BSC DREAM8b dust AOD at 550 nm as well, with a value of ~0.2, which is half of the
sun-photometer-measured AOD at 500 nm, of 0.4. When we scale the BSC-DREAM8b
concentration with these AOD values (multiplying by a factor of 2) the bias is reduced to less
than 10% at 1 km and 50% at 3 km, relative to GARRLiC and LIRIC concentrations. The
GARRLiC and LIRIC mass concentrations are compared also with surface in situ $PM_{10}$
measurements, showing the algorithms overestimating the particle concentration at the surface
level (Fig. 6b, right). We calculate the $PM_{10}$ concentrations from GARRLiC and LIRIC mass
concentrations, as percentages of the particles with diameter less than 10 μm (i.e., 83% and 80%
of the total mass, respectively). Figure 6b (right) shows the GARRLiC and LIRIC $PM_{10}$ surface
values (purple stars in plot), considering marine instead of dust particles at the surface, thus
using the marine particle density for the volume to mass conversion (i.e., $1.7 \, g \, cm^{-3}$ for dry
marine particles (Stock et al., 2011), since the measured RH at the surface is 16%). The




agreement with the surface in situ measurements is better now, but it is only indicatory, since
what we have at the surface is most probably a mixture of marine, continental and dust particles
as shown in Fig 3b.
Summarizing, the GARRLiC and LIRIC retrievals are performing well for the dust episode on
July 26, considering the consistency with the Klett retrievals, the BSC DREAM8b modelled
mass concentration profiles, the surface in situ measurements of the fine mode size distribution,
as well as the expected increase of the dust absorption in the ultraviolet. The discrepancies seen
for the retrieval closer to the surface and the PM$_{10}$ at the surface level can be explained if we
consider the incomplete lidar information in the overlap region.

## 3.2  Marine and polluted continental particle case

On July 15 the lidar measurements at 12:30-14:30 UTC showed a low-AOD layer of non-
depolarizing particles, extending up to 3 km (Fig. 7a). The lack of depolarization indicates
spherical (hydrated) marine particles which is also supported by our source-receptor analysis
(Fig. 7b). Specifically, FLEXPART-WRF simulations show that the particles above Finokalia
station have mainly a marine origin along the whole atmospheric column, with a possible
contribution of continental aerosol from Southern Italy. This scenario is further supported by
AERONET measurements at 13:24 UTC, of low AOD of ~0.06 at 500 nm, high Ångstrom
exponent of ~1.2 at 440/870 nm and low refractive index of ~1.4+i0.0005 at 440 nm, which are
within the climatological value ranges for marine particles and their mixtures, as reported from
Dubovik et al. (2002).
The low AOD is unfavourable for the GARRLiC and AERONET microphysical property
retrievals, especially for the spectral refractive index and SSA (Dubovik et al., 2000b; Lopatin
et al., 2013). The latter require an AOD of at least 0.4 at 440 nm for satisfactory accuracy in
case of sun-photometer-only retrieval (Dubovik et al., 2000b). The lidar information combined
with the sun-photometer measurements in GARRLiC is expected to improve the retrieval in
low AOD cases (Lopatin et al., 2013). Although the AOD requirements have not been
quantified yet for GARRLiC, an AOD of 0.3 at 440 nm is considered sufficient.  As reported
in Dubovik et al. (2002) though, the marine particles rarely exceed the AOD of 0.15 at 440 nm,
thus we do not expect highly accurate refractive index and SSA retrievals from GARRLiC, or
from AERONET/LIRIC, for the marine particles. Even more so, the marine case analysed here
has a much lower AOD, thus we consider the refractive index and SSA retrievals to be only





indicative for this case. In addition, as seen in Fig. 7a, most of the aerosol load is located below
1 km, where the lidar incomplete overlap region is located, which challenges even more the
combined lidar/sun-photometer retrieval.
The GARRLiC and LIRIC retrievals used the lidar measurements at 12:30-14:30 UTC (red box
in Fig. 7a) and the sun-photometer measurements at 13:24 UTC. Figure 8 shows the retrieved
backscatter and extinction coefficients at 355, 532 and 1064 nm, and the corresponding
retrievals from the Klett method. For the latter we consider a LR of 50, 45 and 45, for 355, 532
and 1064, respectively, that closely reproduce the sun-photometer measured AODs of 0.1, 0.05
and 0.02 at 340, 500 and 1020 nm. The agreement between GARRLiC and LIRIC is satisfactory
within the LIRIC uncertainties. Above 550 m, this is also the case for GARRLiC and Klett
backscatter coefficient retrievals, whereas for the extinction coefficients the differences are
within 30% for 355 nm and 10-40% for 532 nm relative to GARRLiC values. In the marine
boundary layer (below 550 m) the Klett NF backscatter and extinction coefficients at 532 nm
show much larger values than the ones retrieved from GARRLiC and LIRIC. This highlights
very vividly the importance of the NF measurements in properly retrieving the marine particle
properties with lidars.
GARRLiC retrieves both fine and coarse particles in this case, which we consider to be mainly
of continental and marine origin, respectively. The fine particle volume size distribution shows
~10% more volume than the AERONET product (also used in LIRIC retrieval), as well as the
surface in situ SMPS measurements (Fig. 9, up left). The SMPS volume size distribution is
converted to $\mu m^3 \mu m^{-2}$ considering that most particles reside from the surface up to ~ 1 km
(Fig. 7). The difference may be partly due to the instrument drying the particle sample, but the
effect is not expected to be that strong since the RH at the surface is 60% and the corresponding
hygroscopic growth is estimated at 1.17 (section 2.1.3, Eq. 2). For the coarse mode, GARRLiC
retrieves ~50% more volume than AERONET. The AERONET SSA and spectral refractive
index retrievals are the same with the GARRLiC fine mode retrievals, or within the retrieval
uncertainty (Fig. 9). These high values of SSA (close to 1) and the refractive index of
1.38±0.4+i0.0005±0.0003 are within the range of climatological values of continental particles,
according to Dubovik et al. (2002). For the GARRLiC coarse mode, the SSA and imaginary
part of the refractive index show very high values for marine particles, which are most probably
false, whereas the real part of the refractive index of ~1.36 agrees well with the climatological
value of 1.36±0.01 for marine particles (Dubovik et al., 2002).





Figure 10a shows the GARRLiC and LIRIC volume concentration profiles, which agree well
within the LIRIC retrieval uncertainties above 550 m, whereas below the GARRLiC
concentration for the coarse particles is larger. Assuming that the marine particles are comprised
only of coarse particles, we derive the marine mass concentration profiles from GARRLiC and
LIRIC as shown in Fig. 10b (left). The mass concentration profiles are calculated from the
coarse volume concentration profiles using a sea salt density of 1.3 g cm$^{-3}$. This value denotes
the density of a sea salt solution at a RH of 50-60 % (Eq. 3 in Zhang et al. (2005)), with the RH
values provided from the RAMS model. Figure 10b (left) shows also the RAMS-ICLAMS sea
salt model mass concentration profile which presents lower values than GARRLiC and LIRIC,
with differences of ~80% and 60% at the surface, respectively. Moreover, GARRLiC and
LIRIC PM$_{10}$ mass concentrations seem to agree well with the surface in situ PM$_{10}$
measurements (Fig. 10b, right), within the time variability of the latter. The GARRLiC and
LIRIC PM$_{10}$ values are calculated using the respective percentages of volume size distributions
for particles with diameter less than 10 μm (i.e., the sum of fine mode volume and 35% of
coarse mode volume for GARRLiC and 50% of total volume for AERONET/LIRIC). The
comparison with the in situ measurements should also consider the drying of the ambient
sample by the in situ instrument. We calculate the GARRLiC and LIRIC "dry" PM$_{10}$,
considering a hygroscopic growth factor of 1.17 at RH=60% at the surface (section 2.1.3). The
"dry" values agree well with the in situ measurements, within the latter time variability.
Summarizing, GARRLiC retrieves more fine particles than AERONET and surface in situ
measurements. The fine particle SSA and refractive index is characteristic of continental
particles. The corresponding coarse mode retrieval probably fails for SSA and the imaginary
part of the refractive index, which are very difficult to be retrieved with low AODs, but the real
part of the refractive index properly assigns the refractive index of marine particles. Both
GARRLiC and LIRIC concentration profiles seem to agree well with the PM$_{10}$ surface in situ
measurements. Since the marine-dominated scenes usually have very low AOD and low vertical
extent (Ho et al., 2015), it is challenging to obtain trustworthy retrievals from GARRLiC and
LIRIC for marine particle scenes. One way to improve the marine retrievals in future efforts
could be to try to increase the lidar information in the overlap region, utilizing for example the
NF lidar measurements, as discussed in section 3.1.1.



## 3.3  Dust and marine case

On July 4 a mixture of dust, marine and continental aerosols was observed at Finokalia station. Figure 11a, shows at 4-6 UTC an advected depolarizing dust plume at 4-6 km and a less-depolarizing plume extending from the ground up to 2-3 km, with volume depolarization ratios at 532 nm of 0.1 and <0.05, respectively. This is a weak dust episode, with a measured column AOD of ~0.15 at 500 nm, which according to the AERONET and GARRLiC uncertainty standards discussed in Section 4.2 should not be sufficient for a full characterization of the particles. The dust and marine particle transport is supported by the BSC DREAM8b dust model and RAMS-ICLAMS sea salt model simulations (Fig. 12b, left), respectively, as well as from our FLEXPART-WRF source-receptor calculations (Fig. 11b). The latter show mainly Saharan dust particles at 4-6 km, marine particles mostly from the Aegean Sea along with continental particles from the Balkans up to 1 km, and a mixture of marine, continental and dust particles at 1-3 km.

GARRLiC retrieves these three layers (Fig. 12a) but it cannot characterize them effectively in terms of their refractive indices, since it is able to retrieve only one refractive index for each mode. For example, the coarse mode of the dust/marine mixture contains dust particles with a real part of refractive index of ~1.55-1.65 (e.g. Kandler et al., 2007) together with marine particles of quite different refractive index, with a real part of ~1.35 (Dubovik et al., 2002). Thus, what we get from GARRLiC as the refractive index of the mixture coarse mode is possibly closer to an average of the refractive indices of dust and marine particles. This is shown in Fig. 13 (down, right), with the GARRLiC coarse mode refractive index to have a value of 1.45 for the real part. The imaginary part of the coarse mode and the SSA show an unusual increase and decrease, respectively, towards the longer wavelengths, which is most probably false. The fine mode should contain mostly continental particles, but the retrieved refractive index of 1.36+i0.001 is more characteristic for marine particles (Dubovik et al., 2002). The AERONET retrieval (used in LIRIC algorithm), assigns a marine refractive index (~1.35+i0.0005) to both fine and coarse particles. The fine mode size distribution compares well with AERONET, but present slightly lower values than SMPS surface in-situ measurements (Fig. 13, up left). With a surface RH of 75%, corresponding to a hygroscopic growth factor of 1.3 (Eq. 2), the GARRLiC fine particle size distribution should be larger than the SMPS dried particle measurements.





Figure 14 shows the potential of GARRLiC to retrieve the "marine" and "dust" components of
the mixture, by changing the definition of the two modes retrieved: instead of "fine" and
"coarse" mode GARRLiC is set to retrieve two modes that span the whole size range so as both
contain coarse particles, and it derives a "dust" mode that contains only coarse particles and a
"marine" mode that contains both fine and coarse particles, of bigger size than "dust". Raptis
et al. (2015) showed similar results for the marine and dust size distribution using their
multimodal analysis for a different dust/marine mixture case during the CHARADMExp
campaign. The retrieved real part of the refractive index is ~1.33 for "marine" particles and
~1.47 for "dust" particles. Although these values are very close to the climatological values for
marine and dust particles, the retrievals of the imaginary part of the refractive index and the
volume concentration profiles are not satisfactory (not shown here). We believe that these
results show a potential for successful marine/dust mixture characterization from GARRLiC in
the future, if the new versions of the algorithm utilize the cross-polarized signals as well. As in
LIRIC, the polarization measurements will help to derive the spherical (marine) and non-
spherical (dust) components of the mixture.
LIRIC provides the dust and marine vertical distribution, since it disentangles the coarse particle
volume concentration profile to its spherical (marine) and non-spherical (dust) components
(Fig. 12a, right). Assuming a very low contribution from dust and marine particles in the fine
mode we acquire the "marine" and "dust" concentration profiles from the spherical and non-
spherical coarse particle concentration profiles, respectively. The left plot of Fig. 12b shows
that LIRIC marine and dust mass concentration profiles have larger values than the BSC
DREAM8b dust and the RAMS-ICLAMS sea salt models, respectively. In order to acquire the
mass concentration profiles, LIRIC dust and marine volume profiles are multiplied with the
density values of $2.6 \, \mathrm{g \, cm^{-3}}$ (Reid et al., 2003) and $1.25 \, \mathrm{g \, cm^{-3}}$, respectively. The marine
particle density corresponds to 60-80% RH (Zhang et al., 2005), as this is provided by the
RAMS model at 0-1 km. We believe that BSC DREAM8b model underestimates the dust
concentration, as for the dust case in section 4.1, since the model AOD of ~0.025 at 500 nm is
approximately 5 times lower than the sun-photometer measured AOD at 550 nm (not taking
into account the AOD contribution of the marine and continental particles). Multiplying the
BSC DREAM8b dust profile by 5 we get a better agreement with LIRIC dust profile at 4-6 km,
but in the mixed layer at 0-3 km this agreement is not satisfactory (not shown here). The RAMS-
ICLAMS model show lower sea salt concentration than LIRIC (as in section 4.2), with ~60 %





differences at the surface level. The right plot in Fig. 12b shows that LIRIC $PM_{10}$ values agree
well with the surface in situ measurements, within the latter time variability. The LIRIC $PM_{10}$
is calculated using the volume percentage of the particles with diameter less than 10 μm (i.e.,
60% of the total volume). Moreover, we calculate the LIRIC "dry" $PM_{10}$ using Eq.1 and
considering a particle hygroscopic growth of 1.3 for RH=75% at the surface (Eq.2). The LIRIC
"dry" $PM_{10}$ is lower than the surface in situ measurements, at ~50% of their mean value. For
GARRLiC the $PM_{10}$ profile cannot be calculated, since the corresponding volume concentration
profile is a mixture of dust, marine and continental particles with unknown density.
Figure 15 shows the backscatter and extinction coefficients retrieved with GARRLiC, LIRIC
and Klett methods. GARRLiC and LIRIC agree well within the LIRIC uncertainties (Fig. 15,
first and second row). The agreement with Klett retrievals is satisfactory for the backscatter
coefficient at 532 and 1064 nm above 550 m, within their uncertainties, with 60-130%
differences seen for the 355 nm retrieval (Fig. 15, third row). As for the marine case in section
4.2, the NF backscatter coefficient at 532 nm show much larger values. The same holds for the
NF extinction coefficient at 532 nm. The Klett extinction coefficients at 1-3 km are up to 60%
and 50% lower than GARRLiC at 355 and 532 nm, respectively.
Overall, this is a challenging case for both GARRLiC and LIRIC algorithms. We can claim that
GARRLiC shows some potential in providing a successful dust and marine microphysical
property characterization in case more information (e.g. cross-polarized lidar signal) is included
in the retrieval. Moreover, the LIRIC capability of providing the vertical distribution of dust
and marine particles is mostly successful, comparing the results with our source-receptor
simulations and the surface in situ $PM_{10}$ measurements. As is the case also for the marine
particle characterization in section 4.2, we believe that this retrieval will be greatly benefited
from NF measurements.
**4   Summary and Conclusions**
GARRLiC and LIRIC algorithms provide the great innovation of retrieving the vertical
distribution of aerosol microphysics utilizing the synergy of the elastic backscatter lidar and
sun-photometer techniques. This way, the algorithms show the potential to effectively
characterize the vertical distribution of fine, coarse spherical and coarse non-spherical particle
concentrations in the case of LIRIC, and the concentration profiles of fine and coarse particles,




along with their column-averaged size, shape and spectral refractive index, in case of
GARRLiC.
In this study we used both algorithms to characterize three cases of dust and marine presence
during the ESA-CHARADMExp experimental campaign. For the first case GARRLiC achieves
a successful retrieval of the dust vertical distribution and microphysical characterization that
agrees well with AERONET and climatological values for dust. Both LIRIC and GARRLiC
concentration profiles are found to be consistent with the BSC DREAM8b dust vertical
structure, showing though larger values from the surface in situ $PM_{10}$ measurements. For the
second case consisting of mainly marine particles, both algorithms provide satisfactory
concentration retrievals comparing with the surface in situ $PM_{10}$ measurements. The GARRLiC
microphysical property retrieval is mostly not successful for the marine particles. This is due to
the difficulties posed by the really low AOD and the insufficient lidar information in the overlap
region, where most of the marine aerosol load resides. Last, for the more challenging case of
dust and marine mixture, LIRIC provides the dust and marine particle vertical structure due to
its capability to retrieve the coarse mode spherical (marine) and non-spherical (dust)
components. GARRLiC shows potential in disentangling the marine and dust components, if
more information is included in the algorithm input.
The difficulties posed in retrieving the concentration profiles and the microphysical properties
of dust and marine particle mixtures in the atmospheric column have to do with the low AOD
of the marine plumes, the insufficient lidar information in the overlap region and the number of
modes considered from the retrievals. For GARRLiC, the retrieval of multiple modes would be
possibly feasible in the future with the incorporation of polarimetric measurements from the
sun-photometer and/or the cross-polarized and Raman signals from the lidar. Moreover, we
could try to increase the near-to-surface information from the lidar, performing the signal gluing
technique between the FF and NF measurements. We aim to continue investigating the
GARRLiC and LIRIC potential for aerosol characterization and follow related improvements
in the framework of the ACTRIS-2 project and the experimental campaigns that are dedicated
to that objective.



**Acknowledgements**
The research leading to these results has received funding from the European Union's Horizon
2020 Research and Innovation Programme ACTRIS-2 (grant agreement no. 654109). The work
has been developed under the auspices of the ESA-ESTEC project "Characterization of Aerosol
mixtures of Dust And Marine origin" contract no. IPL-PSO/FF/lf/14.489. The publication was
supported by the European Union's Horizon 2020 Research and Innovation programme under
grant agreement No 602014, project ECARS (East European Centre for Atmospheric Remote
Sensing). BSC-DREAM8b simulations were performed on the Mare Nostrum supercomputer
hosted by the Barcelona Supercomputing Center–Centro Nacional de Supercomputación
(BSC).



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





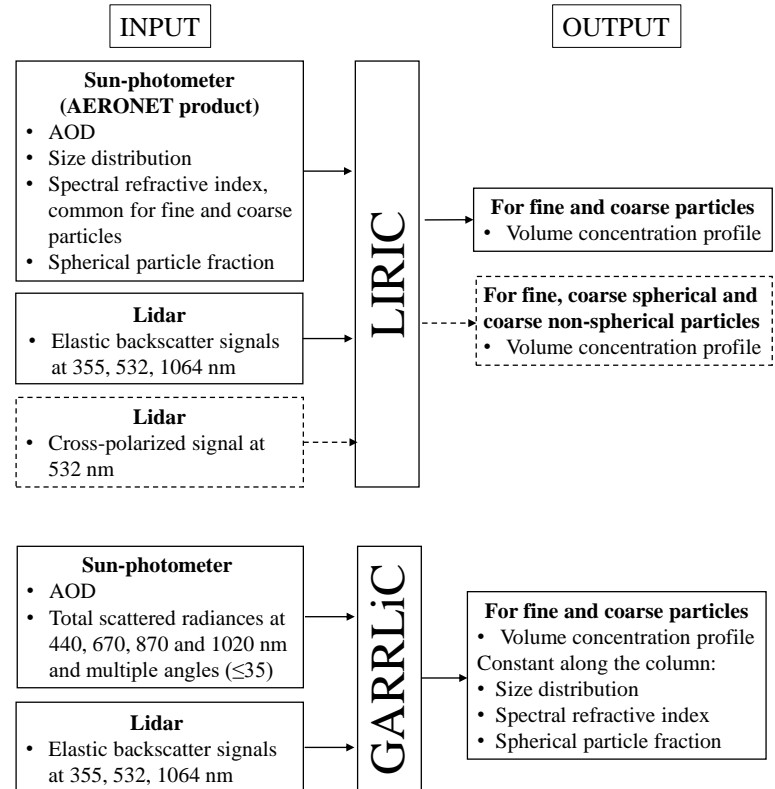

3    Figure 1. GARRLiC and LIRIC algorithm input and output parameters. For LIRIC, the output

4    in case of using the cross-polarized signal at 532 nm is shown in the dashed box.



1    a) 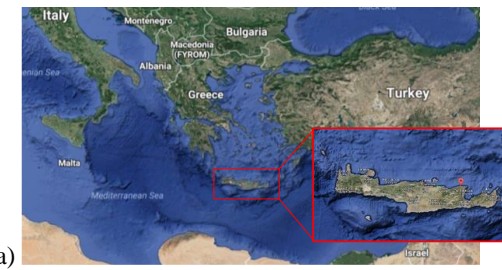     b) 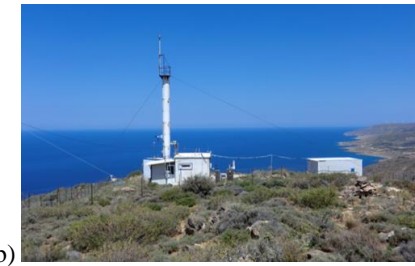

3    Figure 2. a) Location of Finokalia station (red dot) in Crete island, Greece. b) Sea view from

4    the station.





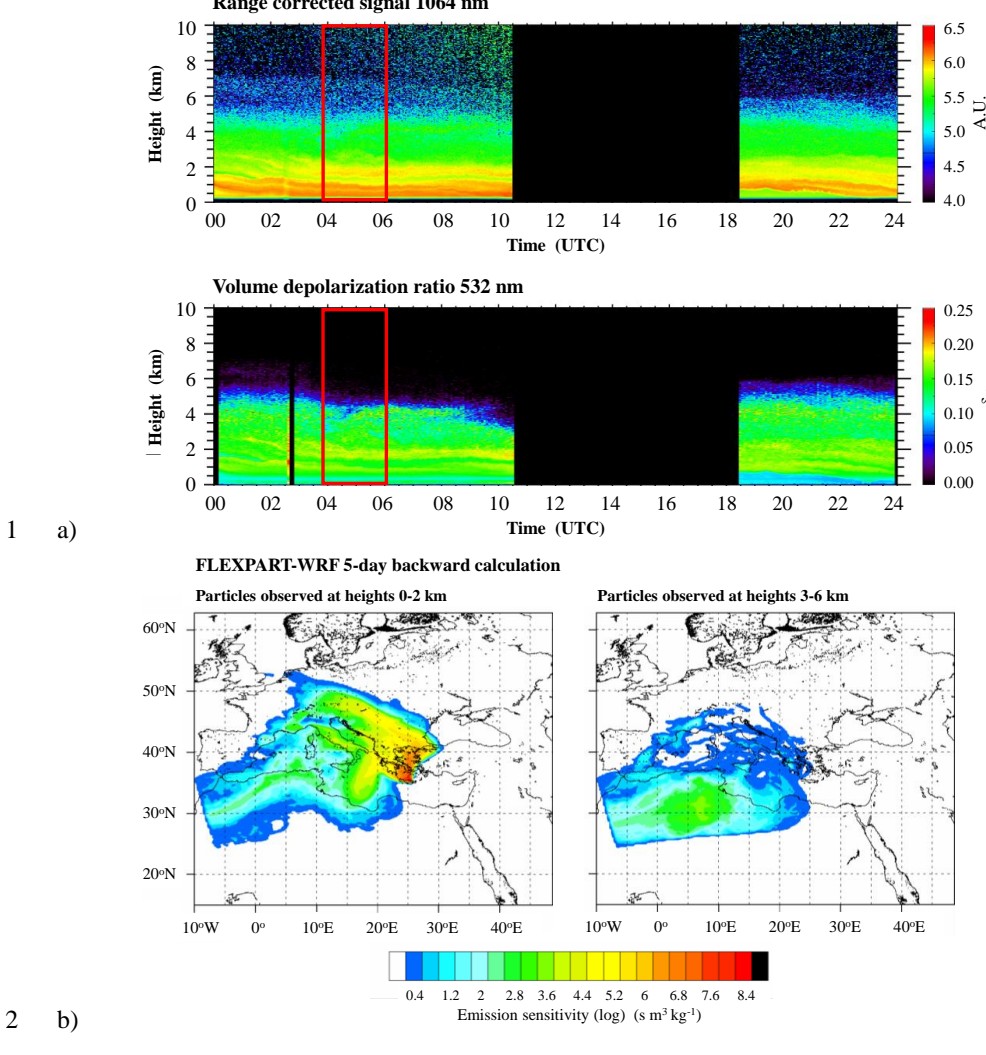

a)
b)

Figure 3. a) Range-corrected backscattered signal at 1064 nm in arbitrary units (top) and volume

depolarization ratio at 532 nm (bottom) from Polly[XT] OCEANET lidar, at Finokalia, Crete, on

June 26, 2014. The red rectangle indicates the time range of the measurements used for

GARRLiC and LIRIC retrievals (04:00-06:00 UTC). b) Five day backward FLEXPART-WRF

calculation of emission sensitivity (i.e., residence time in the lowest 1 km in the atmosphere) in

$\log(s\ m^3\ kg^{-1})$ for the particles arriving at 0-2 km (left) and 3-6 km (right) at 04:00 UTC.











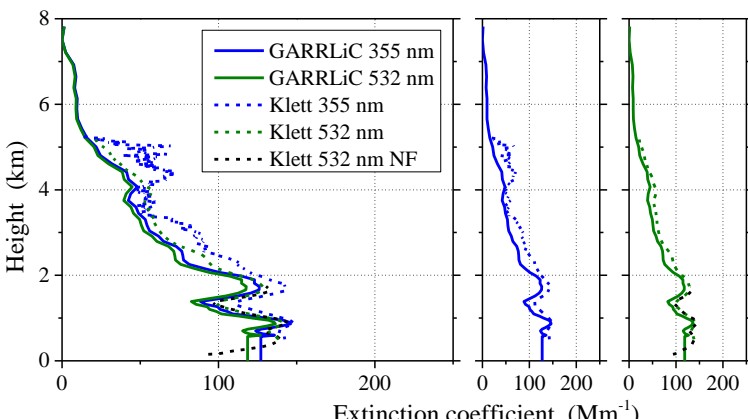

Figure 4: Backscatter and extinction coefficient retrievals, at Finokalia, Crete, on June 26, 2014,
at 04:00-06:00 UTC. First and second rows: Backscatter and extinction coefficients from
GARRLiC and LIRIC. Third and fourth rows: Backscatter and extinction coefficients from
GARRLiC and Klett. In each row the first plot contains the results for all wavelengths (i.e.,
355, 532 and 1064 nm) and the next three plots contain the results for each wavelength
separately.





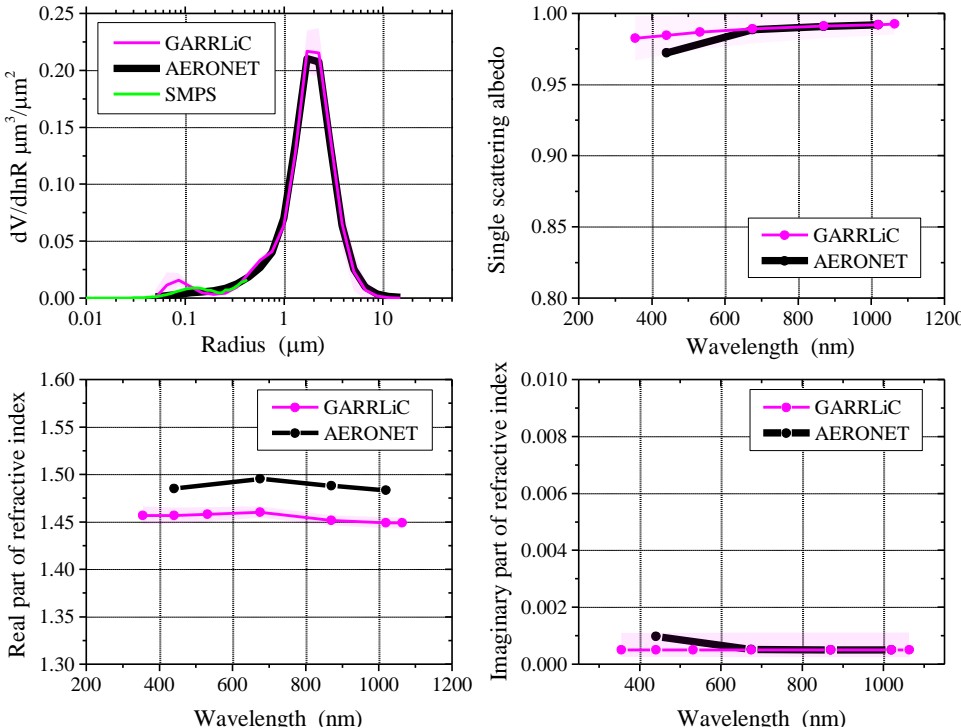

Figure 5: GARRLiC retrievals (pink) of size distribution (up-left), spectral SSA (up-right),
spectral real and imaginary part of the refractive index (bottom –left and right), on June 26,
2014, at 04:00-06:00 UTC, in Finokalia, Crete. The black line shows the AERONET retrieval
at 04:54 UTC (used also in LIRIC). The green line in the size distribution plot (up-left) is the
mean value of the surface in situ SMPS measurements at 04:00-06:00 UTC.





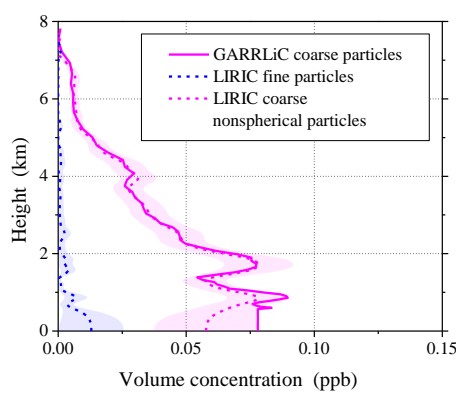

1    a)

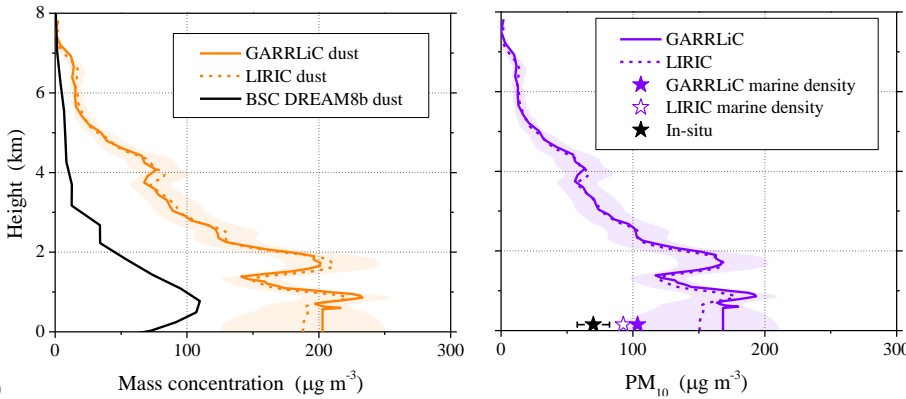

2    b)

Figure 6: a) Volume concentration profiles for GARRLiC coarse particles (pink), and LIRIC

fine (dash blue) and coarse nonspherical particles (dash pink), on June 26, 2014, at 04:00-06:00

UTC, in Finokalia, Crete. b) Left: Dust mass concentration profiles from GARRLiC (orange),

LIRIC (dash orange) and BSC DREAM8b model (black) (the latter at 05:00 UTC). Right: $PM_{10}$

profiles from GARRLiC (purple) and LIRIC (dash purple), along with their surface values,

considering only marine particles at the surface ("GARRLiC marine density" and "LIRIC

marine density" denoted by purple star and white star, respectively). The black star denotes the

surface in situ measurements at 05:00-06:00 UTC (mean and time variability).





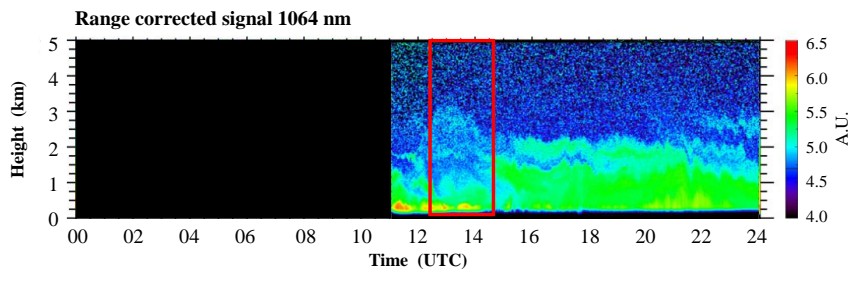

a)

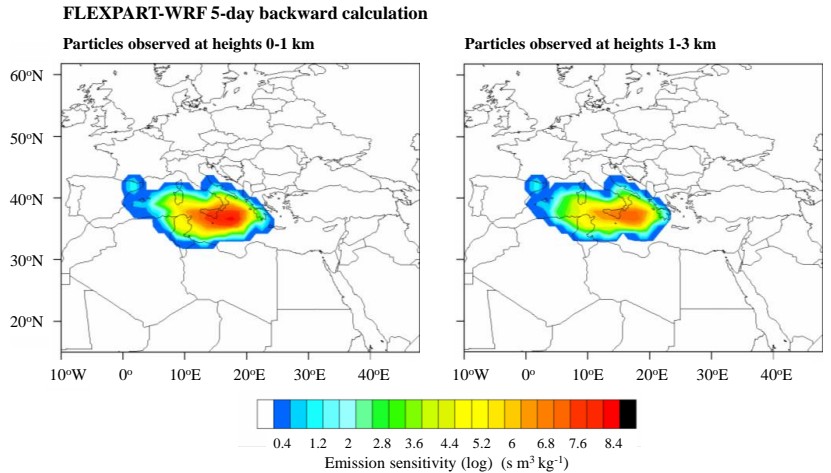

b)
Figure 7. a) Range-corrected backscattered signal at 1064 nm in arbitrary units from Polly[XT]
OCEANET lidar, at Finokalia, Crete, on July 15, 2014. The red rectangle indicates the time
range of the measurements used for the GARRLiC and LIRIC retrievals (12:30-14:30 UTC).
b) Five day backward FLEXPART-WRF calculation of emission sensitivity (i.e., residence
time in the lowest 1 km in the atmosphere) in $\log(s\,m^3\,kg^{-1})$ for the particles arriving at the
layers 0-1 km (left) and 1-3 km (right) at 14:00 UTC.









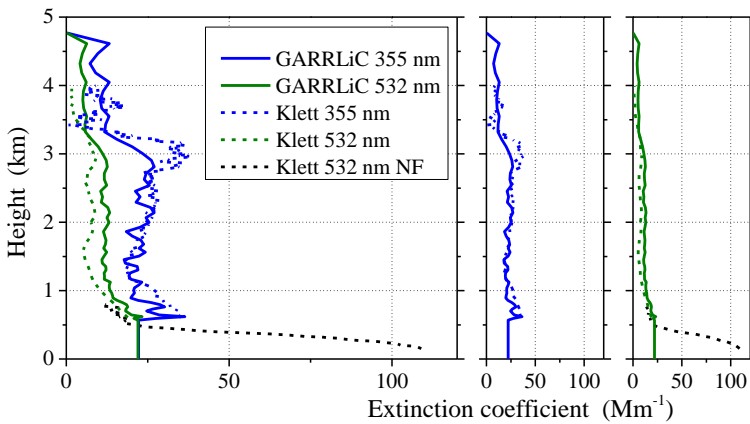

3    Figure 8: As in Fig. 4 for backscatter and extinction coefficient retrievals at Finokalia, Crete,

4    on July 15, 2014, at 12:30-14:30 UTC.





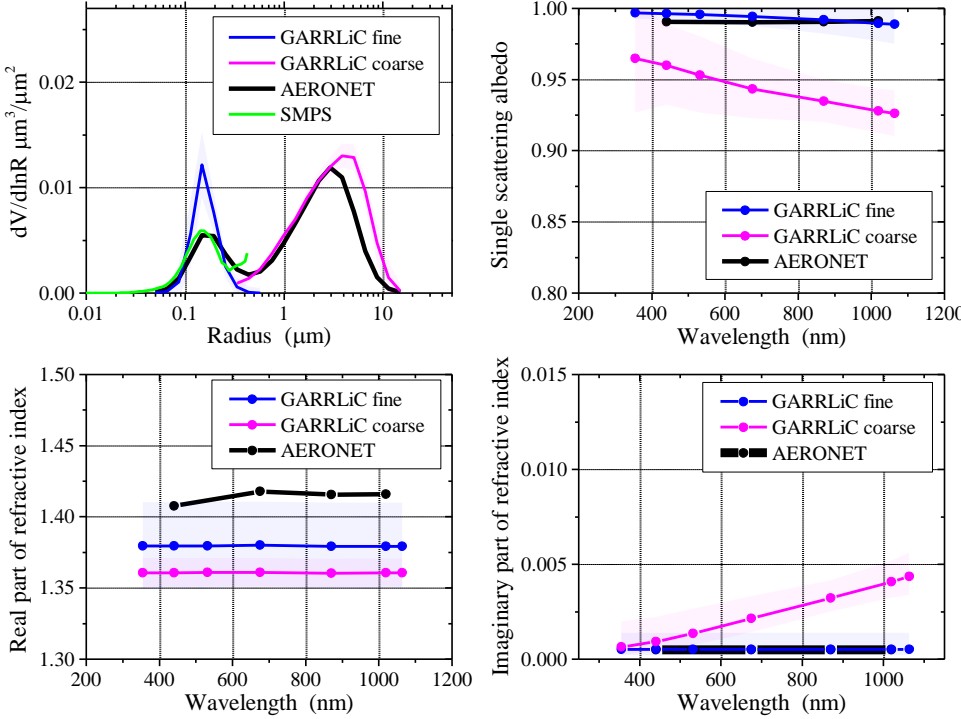

Figure 9: GARRLiC retrievals for fine (blue) and coarse particles (pink) of size distribution
(up-left), spectral SSA (up-right), spectral real and imaginary part of the refractive index
(bottom –left and right), at Finokalia, Crete, on July 15, 2014, at 12:30-14:30 UTC. The black
line shows the AERONET retrieval at 13:24 UTC. The green line in the size distribution plot
(up-left) is the mean value of the surface in situ SMPS measurements at 12:00-13:20 UTC.





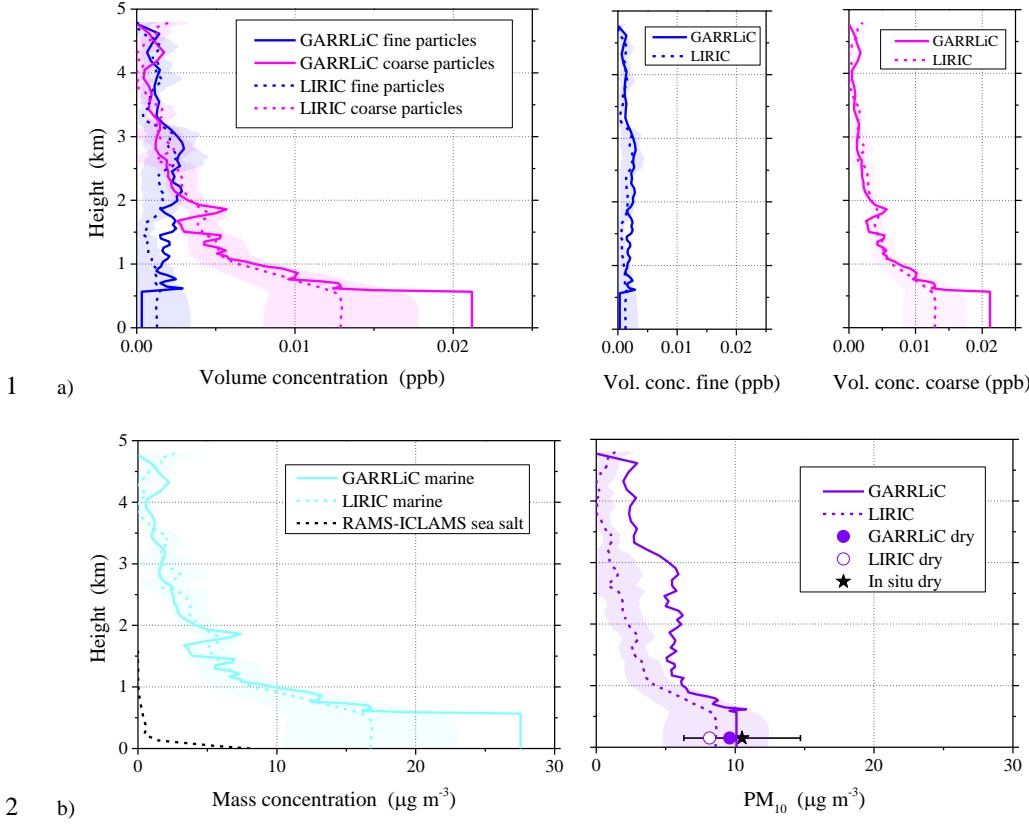

Figure 10: a) Volume concentration profiles for GARRLiC fine (blue) and coarse particles
(pink) and LIRIC fine (dash blue) and coarse spherical particles (dash pink), at Finokalia, Crete,
on July 15, 2014, at 12:30-14:30 UTC. b) Left: GARRLiC (light blue) and LIRIC (dash light
blue) marine particle mass concentration profiles, along with the RAMS-ICLAMS sea salt mass
concentration profile (black) at 13:00 UTC. Right: $PM_{10}$ profiles from GARRLiC (purple) and
LIRIC (dash purple), along with the "dry" GARRLiC and LIRIC $PM_{10}$ at the surface (purple
and white circles, respectively). The black star denotes the in situ $PM_{10}$ measurements at 4-5
UTC (mean and time variability).





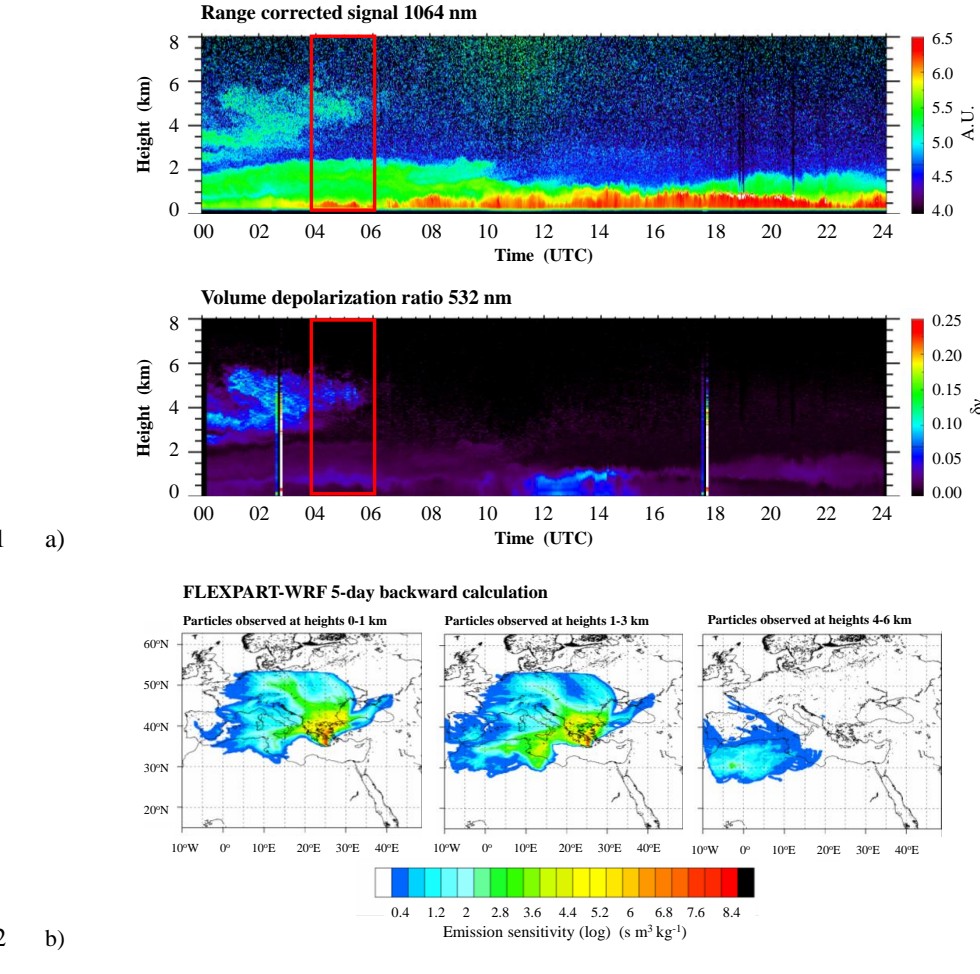

1    a)

2    b)

Figure 11: a) Range-corrected backscattered signal at 1064 nm in arbitrary units (top) and volume depolarization ratio at 532 nm (bottom) from Polly[XT] OCEANET lidar, at Finokalia, Crete, on July 4, 2014. The red rectangle indicates the GARRLiC and LIRIC retrievals (04:00-06:00 UTC). b) Five day backward FLEXPART-WRF calculation of emission sensitivity (i.e., residence time in the lowest 1km in the atmosphere) in $\log(s\ m^3\ kg^{-1})$ for the particles arriving at  heights 0-1 km (left), 1-3 km (middle) and 4-6 km (right), at 07:00 UTC.





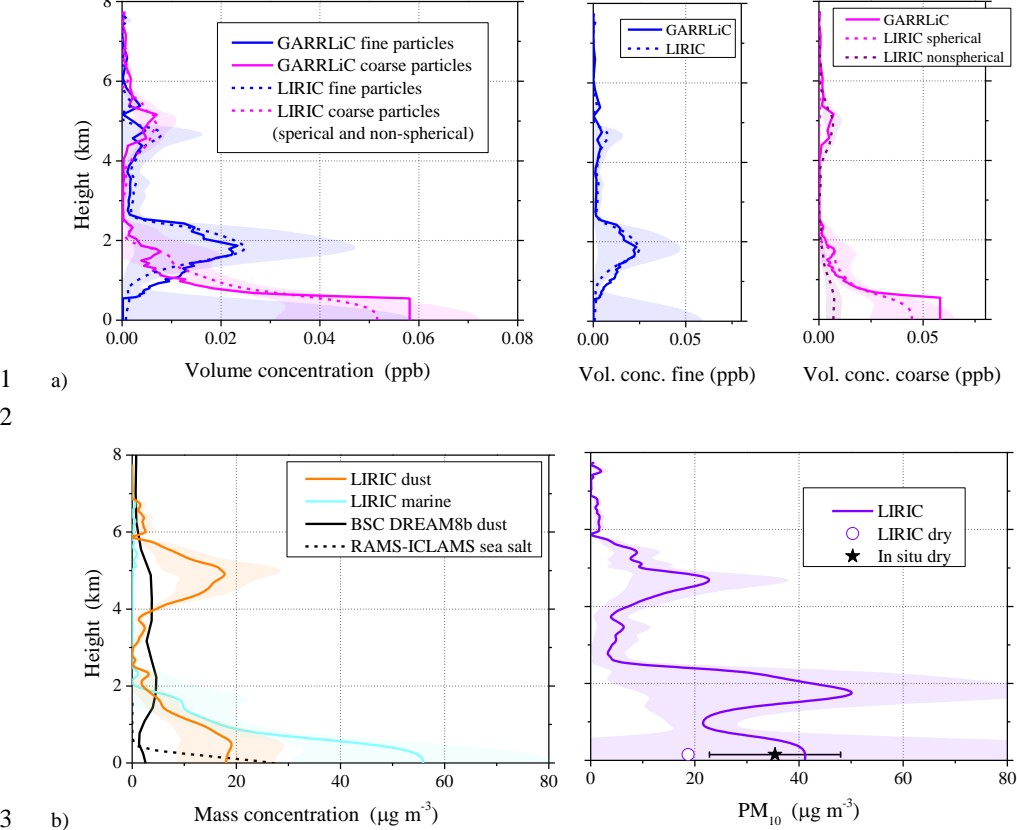

a)
b)
Figure 12: a) Left: Volume concentration profiles for GARRLiC fine (blue) and coarse particles
(pink) and LIRIC fine (dash blue) and total coarse particles (dash pink), at Finokalia, Crete, on
July 4, 2014, at 04:00-06:00 UTC. Middle: Volume concentration of fine particles from
GARRLiC (blue) and LIRIC (dash blue). Right: Volume concentration of coarse particles from
GARRLiC (pink) and LIRIC, disentangled in the spherical (dash pink) and non-spherical (dash
purple) components. b) Left: Mass concentration profiles for LIRIC dust (orange) and marine
particles (light blue), along with the modelled dust (black) and sea salt (dash black) particle
concentration profiles (both at 05:00 UTC). Right: PM$_{10}$ profile from LIRIC (purple), along
with the "dry" LIRIC PM$_{10}$ at the surface (white circle). The black star denotes the surface in
situ PM$_{10}$ measurements at 4-5 UTC (mean and time variability).





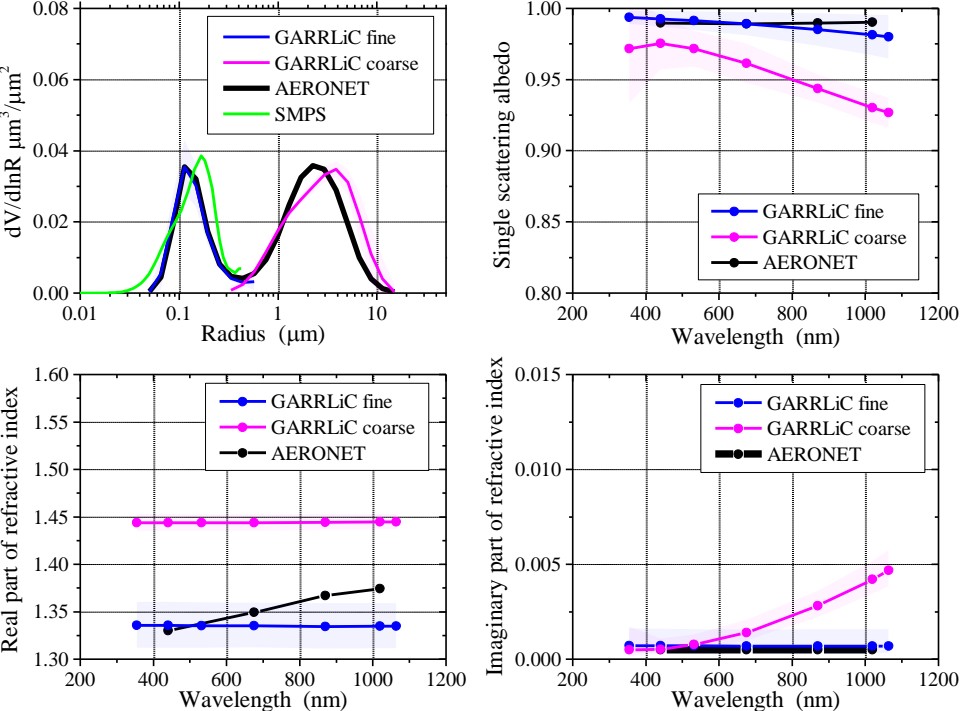

Figure 13: GARRLiC retrievals for fine (blue) and coarse particles (pink) of size distribution
(up-left), spectral SSA (up-right), spectral real and imaginary part of the refractive index
(bottom –left and right), at Finokalia, Crete, on July 4, 2014, at 04:00-06:00 UTC. The black
line shows the AERONET retrieval at 05:49 UTC. The green line in the size distribution plot
(up-left) is the mean value of the surface in situ SMPS measurements at 04:00-06:00 UTC.



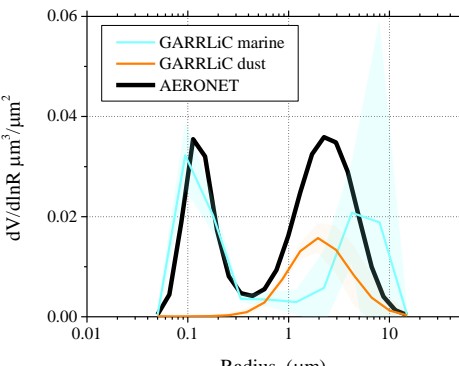
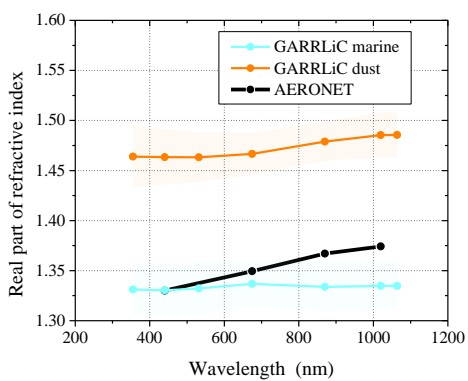

Figure 14: Potential of GARRLiC to retrieve "marine" (light blue) and "dust" particle (orange)
size distribution (left) and spectral real part of the refractive index (right). The retrieval refers
to measurements at Finokalia, Crete, on July 4, 2014, at 04:00-06:00 UTC. The black line shows
the AERONET retrieval at 05:49 UTC.











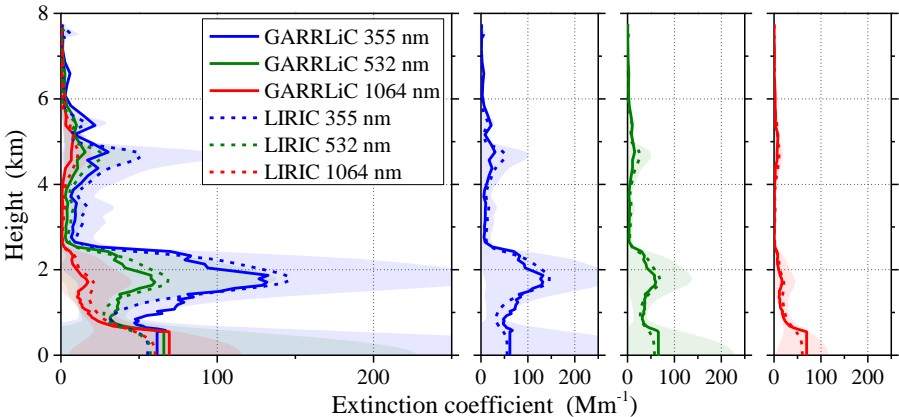

3   Figure 15: As in Fig. 4 for backscatter and extinction coefficient retrievals at Finokalia, Crete,

4   on July 4, 2014, at 04:00-06:00 UTC.