# Peer review of "GARRLiC and LIRIC: strengths and limitations for the characterization of dust and marine particles along with their mixtures"

_Atmospheric Measurement Techniques, 2017_

## Referee Comment (RC1) · Anonymous Referee #1 · 24 Jul 2017

The authors provide comparison the inversion of lidar data combined with sun photometer (SP) measurements using GARRLIC and LIRIC algorithms. These algorithms are widely used in the lidar community, so their comparison is important. Moreover, inversion of lidar observations collected during CHARADMExp helps to understand better the potential and issues of lidar-SP combining. The manuscript is well written, the authors understand the limitations of their approach and openly discuss it. I think manuscript can be published after minor revisions.

I think in the introduction it would be useful to mention the main (to my opinion) issue of lidar-SP combining. The modal radii of both modes are taken from SP and assumed

to be height independent (refractive index as well). Still these values may change with height, for example, due to hygroscopic growth, or due to the presence of layers with different aerosol types. So what will be impact of this height variation to the retrieval?

Additional comments

1. Reference "Müller, et al., Atmos. Meas. Tech. Discuss., 8, , 2015". Why AMTD? Wasn't it published?

2. p.11 ln.20 "More specifically, they managed to reproduce this backscatter spectral dependence with imaginary part values of 0.005-0.05 at 355 nm and 0.005 at 532 nm".

In the paper Veselovskii et al., 2016, the simulation was performed imaginary part at 355 nm (mI(355)) varying in the range 0.005-0.05, but values of BAE close to experimentally observed were obtained for mI(355) about 0.01.

3. p.11, ln.22 "Although these values are not the same with the 22 retrieved 0.001 at 355 nm and 0.0005 at 532 nm for our case..."

These values of mI are too low for dust

4. p.11, ln.23 "The backscatter spectral dependence can be a combination of the effect that different factors have on the backscattered light, as the size, shape or orientation of the dust particles" I think this statement is unclear and unsupported. Yes, size distribution will influence". I am not sure about shape, at least not in the frame of spheroids model. How orientation can influence?

5. p.11, ln.27. "Differences are seen only for the real part of the refractive index, which for GARRLiC is at ∼1.45, at the low end of the dust climatological value range of 1.48±0.05-1.56±0.03 as reported in Dubovik et al. (2002)." AERONET can't be used as a referencel value for dust refractive index, because it may underestimate the real part. Laboratory and in situ measurements are more reliable.

6. Fig.5. AERONET shows increase of mI at short wavelengths, which agrees with

known in situ measurements, while mI in GARRLIC doesn't show spectral dependence. Can you comment it? Low values of mI are usually obtained in inversion when high depolarization ratios are considered, because spheroids model can reproduce it only for low mI. Do authors use depolarization ratio in retrievals?

7. Fig.8. Second row. Misprint. "Garrlic 532" is printed twice

---

## Referee Comment (RC2) · Anonymous Referee #2 · 24 Jul 2017

Tsekeri et al. present a study on the performance of two lidar algorithms in characterizing the vertical distribution of Saharan dust, marine particles and mixtures thereof for three case studies.

The evaluation of retrieval algorithms that combine lidar with passive remote sensing measurements, such as sunphotomers or polarimeters, is an important piece in a much larger puzzle that aims to understand how to effectively combine measurements obtained from a number of different instruments in order to maximize the information content available in the data to accurately retrieve optical and physical properties of aerosols, allowing us to have a complete spatial (horizontal and vertical) and temporal

characterization of the atmosphere.

The manuscript is well written and well structured, and the methodology used is sound. However, the results are presented mostly in qualitative terms. The readers would benefit from having more quantitative results described in the paper. I recommend the publication of this manuscript once the authors address the following points:

Specific comments:

Page 3: lines 26-28: This segment does not read well. Perhaps change to something like: "(. . .) coarse particles. The cross-polarized lidar signal at 532 nm allows the decoupling of the coarse mode into its spherical and non-spherical components"

Page 4, line 7-9: Please elaborate a bit more on the retrieval uncertainties. How do you determine them? What's the difference in determining the uncertainties for the total-column microphysical retrievals vs. for the profile retrieval of concentrations?

Page 4, line 18: What do you mean by "whereas otherwise" in this case? It doesn't seem to fit in the sentence.

Page 5, line 25-26: Suggestion: "Dust transport, while less frequent during the dry period, it is still observed (e.g. . . .) and it is characterized by a transport pattern (. . .)"

Page 7, line 1: "instrument and calibration precision" instead of "instrument precision and calibration precision".

Page 7, line 2: replace "Visible" by "visible range".

Page 7, line 16: replace "extent" by "fraction".

Page 8, line 22: Why 40,000? Is that an arbitrary or standard number of particles that modelers use, or was there another reason for that choice?

Page 9, line 12: what is Eta in "24 Eta vertical layers"?

Page 9, line 13: Change 1/3 to 0.33

Page 10, lines 3-9: I think it might be good to include a shorter version of this in the abstract since currently you do not mention anything about the models or in situ measurements that you use to aid in the characterization study.

Page 10, line 28: How much is "quite well" in % difference?

Figure 4: Consider labeling each panel like 4a, 4b, etc so you don't have to refer to them as "first and second row in Fig 4".

Page 11, line 1: replace "cut above" by "restricted to"

Page 12, line 7-8: "Moreover, due to the low RH at the surface (16%) we do not (...)"

Page 12, line 10: What's excellent in % difference? Please quantify it.

Figure 6: Instead of having Fig 6b with two panels, just relabel each plot in Figure 6 as 6a, 6b, 6c. Makes it easier to reference.

Page 13, line 1: replace "indicatory" by "indicative" or "qualitative"

Page 14, line 7: missing LR unit (sr).

Page 19, line 3-17: Please quantify the agreement levels that you mention in this paragraph. How many %?

Page 19, line 8: replace "from" by "than"

Page 19, line 24: replace "near-to-surface" by "near-surface".

---

## Referee Comment (RC3) · Anonymous Referee #3 · 25 Jul 2017

Both active and passive remote sensing instruments have their own advantages and disadvantages in terms of the vertically-resolved retrievals of the aerosol microphysical parameters. It is clear that the combination of active/passive technologies is absolutely necessary in order to improve the quality of retrievals. Alexandra Tsekeri and co-authors are moving in that direction.

Finokalia lidar station is famous to have a unique geographical location allowing observation of the different combinations of the aerosol particles from Central Europe and Africa/Sahara. It is nice to get a reminder that Finokalia station is operational and actively delivering the valuable scientific data related to the dust and marine aerosols.

The paper in overall is very well written. It is pleasure to read the text. The wording and grammar are close to be perfect.

I recommend to accept this manuscript for the publication after a few minor corrections.

General comments:

1. After reading the paper I got an impression that there is some kind of luck that GARRLiC and LIRIC algorithms work and result in a reasonable aerosol microphysical retrievals. Here is the list of introduced assumptions to support this impression:

A) "The volume concentration below the lowest height of the lidar signals is considered to be constant" (page 4, line 9)

It is very difficult to imagine the constant volume concentration in the first few hundreds meters above the ground. Please keep in mind that the comparisons with in situ are performed using the near-surface data.

B) The Raman signals from the lidar are such a useful piece of information about aerosols, but not used in the retrievals at all (see Fig. 1). There is only a plan to use the extinction optical coefficients in future.

C) "GARRLiC ... is able to retrieve only one refractive index for each mode." (page 16, line 14) "considering refractive index to be constant along the atmospheric column" (page 4, line 6)

The assumption of only one refractive index per profile is very damaging for the whole idea of vertically-resolved microphysical retrievals. The aerosols of different nature (thus, different refractive indexes) are often reside in a different layers of the same vertical profile. Please consider in your future studies to eliminate this assumption.

D) Some additional difficulties on the top of that:

- "In addition, as seen in Fig. 7a, most of the aerosol load is located below 1 km, where the lidar incomplete overlap region is located, which challenges even more the

combined lidar/sun-photometer retrieval." (page 14, line 1)

As a result, there is a nice agreement between GARRLiC and LIRIC in terms of volume concentration (see Fig. 10a), very strong disagreement between GARRLiC and LIRIC in terms of ambient PM10 (up to 3-4 times or so, see Fig. 10b.right above 1 km), and than finally fair agreement between GARRLiC, LIRIC, and in situ in terms of dry PM10 (see Fig. 10b.right below 1 km). All these sudden turns are really thrilling! Please provide some explanation in the text.

2. GARRLiC and LIRIC algorithms are based on the usage of pre-calculated AERONET products. Without going too much into a details, they sound almost like a twins or, at least, share some part of the software code. For the final users at the lidar stations it is not convenient to have several twins-like algorithms in a package. It is confusing to have similar results for the one group of microphysical parameters and different results for the other group of parameters. Is there a plan to come up with the best way on how to combine the lidar/sun-photometer data that will merge/replace GARRLiC and LIRIC in a single algorithm? It is highly desirable if authors will share their vision regarding this issue in a paragraph of text.

Specific comments:

1. Page 4. Line 4: "The algorithm calculates the size distribution, spherical particle fraction and spectral complex refractive index, separately for fine and coarse particles, considering them constant along the atmospheric column, and the volume concentration profiles of fine and coarse particles."

This sentence is quite unclear and ambiguous. Please consider to split it into two simpler sentences.

2. Page 11. Line 11

"m." instead of "m:"

---

## Author Comment (AC1) · 21 Aug 2017

The authors provide comparison the inversion of lidar data combined with sun photometer (SP) measurements using GARRLIC and LIRIC algorithms. These algorithms are widely used in the lidar community, so their comparison is important. Moreover inversion of lidar observations collected during CHARADMExp helps to understand better the potential and issues of lidar-SP combining. The manuscript is well written, the authors understand the limitations of their approach and openly discuss it. I think manuscript can be published after minor revisions.

REPLY: We thank the reviewer for his/her kind words!

I think in the introduction it would be useful to mention the main (to my opinion) issue of lidar-SP combining. The modal radii of both modes are taken from SP and assumed to be height independent (refractive index as well). Still these values may change with height, for example, due to hygroscopic growth, or due to the presence of layers with different aerosol types. So what will be impact of this height variation to the retrieval?

REPLY: We agree with this comment. We added in the text (pg. 4, lines 18-20): "In case of multi-mode aerosol mixtures and/or change of microphysical properties with height due to particle hygroscopic growth (e.g. Tsekeri et al., 2017) an inherent deficiency of both algorithms is the number of aerosol modes retrieved...". Concerning the impact of this effect, this should be the subject of a different study, which will compare GARRLiC and LIRIC retrievals with height-resolved retrievals. In any case, we already mention in the same paragraph (pg. 4, lines 24-28): "Both algorithms work well for individual aerosol components or mixtures of (mainly) fine (e.g. pollution) and (mainly) coarse (e.g. dust) particles, but they should not be able to fully characterize the mixture components in case of more than one fine or coarse mode in the mixture, as in smoke/pollution or dust/marine mixture cases."

Additional comments

1. Reference "Müller, et al., Atmos. Meas. Tech. Discuss., 8, , 2015". Why AMTD? Wasn't it published?

REPLY: We changed it in Müller et al., 2016.

2. p.11 ln.20 "More specifically, they managed to reproduce this backscatter spectral dependence with imaginary part values of 0.005-0.05 at 355 nm and 0.005 at 532 nm". In the paper Veselovskii et al., 2016, the simulation was performed imaginary part at 355 nm (ml(355)) varying in the range 0.005-0.05, but values of BAE close to experimentally observed were obtained for ml(355) about 0.01.

REPLY: We corrected the text as following (pg. 11, lines 22-24): "More specifically,
they managed to reproduce this backscatter spectral dependence with imaginary part values of  ${\sim}0.01$  at 355 nm and 0.005 at 532 nm."

3. p.11, ln.22 "Although these values are not the same with the retrieved 0.001 at 355 nm and 0.0005 at 532 nm for our case: : :" These values of ml are too low for dust

REPLY: We inserted the following explanation in the following paragraph (pg. 12, lines 2-5): "The same is true for the low values of the imaginary part, due to the mixture of dust with imaginary part of e.g. 0.05 at 532 nm (e.g. Wagner et al., 2012) and marine particles with imaginary part of  $\sim$ 0.0005 at 532 nm (e.g. Babin et al., 2003)."

4. p.11, ln.23 "The backscatter spectral dependence can be a combination of the effect that different factors have on the backscattered light, as the size, shape or orientation of the dust particles" I think this statement is unclear and unsupported. Yes, size distribution will influence". I am not sure about shape, at least not in the frame of spheroids model. How orientation can influence?

REPLY: We agree that the orientation influence of the backscatter spectral dependence is a speculation that needs to be investigated further. We deleted it form the text.

5. p.11, ln.27. "Differences are seen only for the real part of the refractive index, which for GARRLiC is at  $\hat{a}$ Lij1.45, at the low end of the dust climatological value range of  $1.48\pm0.05$ - $1.56\pm0.03$  as reported in Dubovik et al. (2002)." AERONET can't be used as a reference value for dust refractive index, because it may underestimate the real part. Laboratory and in situ measurements are more reliable.

REPLY: We agree. That's why we continue with the following statement in the text: "This value though is much lower than expected for dust from West Sahara in situ measurements, reporting values of 1.55-1.65 (e.g. Kandler et al., 2007), and it may be due to the marine particle mixture at lower heights, with real part of refractive index of  $\sim$ 1.35."

6. Fig.5. AERONET shows increase of mI at short wavelengths, which agrees with
known in situ measurements, while mI in GARRLIC doesn't show spectral dependence. Can you comment it? Low values of mI are usually obtained in inversion when high depolarization ratios are considered, because spheroids model can reproduce it only for low mI. Do authors use depolarization ratio in retrievals?

REPLY: The increase shown in mI from AERONET is within the GARRLiC mI retrieval uncertainty. We haven't included the depolarization ratios in the retrievals, since GAR-RLiC does not use (yet) the depolarization ratio as input for the retrieval.

7. Fig.8. Second row. Misprint. "Garrlic 532" is printed twice

REPLY: We corrected it in the Figure.

---

## Author Comment (AC2) · 21 Aug 2017

Tsekeri et al. present a study on the performance of two lidar algorithms in characterizing the vertical distribution of Saharan dust, marine particles and mixtures thereof for three case studies. The evaluation of retrieval algorithms that combine lidar with passive remote sensing measurements, such as sunphotomers or polarimeters, is an important piece in a much larger puzzle that aims to understand how to effectively combine measurements obtained from a number of different instruments in order to maximize the information content available in the data to accurately retrieve optical and physical properties of aerosols, allowing us to have a complete spatial (horizontal and vertical) and temporal characterization of the atmosphere. The manuscript is well written and well structured, and the methodology used is sound. However, the results are presented mostly in qualitative terms. The readers would benefit from having more quantitative results described in the paper. I recommend the publication of this manuscript once the authors address the following points:

REPLY: We thank the reviewer for his/her kind words! We tried to address all his/her points, as shown below.

Specific comments:

Page 3: lines 26-28: This segment does not read well. Perhaps change to something like: "(...) coarse particles. The cross-polarized lidar signal at 532 nm allows the decoupling of the coarse mode into its spherical and non-spherical components"

REPLY: We inserted the change in the text.

Page 4, line 7-9: Please elaborate a bit more on the retrieval uncertainties. How do you determine them? What's the difference in determining the uncertainties for the total-column microphysical retrievals vs. for the profile retrieval of concentrations?

REPLY: The retrieval uncertainties of the total-column microphysical parameters have been developed following the approach described by Dubovik et al. (2000), whereas the profile retrieval uncertainties are currently under development. For the latter we need to take into account the lidar measurements as well. We changed the text as following (pg. 4, lines 13-16): "The retrieval uncertainties of the microphysical parameters are provided as well, following the approach described by Dubovik et al. (2000) and the profile retrieval uncertainties are currently under development."

Page 4, line 18: What do you mean by "whereas otherwise" in this case? It doesn't seem to fit in the sentence.

REPLY: We mean that it retrieves 3 modes only for the concentration profile, whereas otherwise, for all other microphysical properties it retrieves one mode.

AMTD
Page 5, line 25-26: Suggestion: "Dust transport, while less frequent during the dry period, it is still observed (e.g. ...) and it is characterized by a transport pattern (...)"

REPLY: We changed the text accordingly.

Page 7, line 1: "instrument and calibration precision" instead of "instrument precision and calibration precision".

REPLY: We changed the text accordingly.

Page 7, line 2: replace "Visible" by "visible range".

REPLY: We changed the text accordingly.

Page 7, line 16: replace "extent" by "fraction".

REPLY: It is the height extend and not the fraction of the fine particles. We changed the text as following (pg. 7, lines 20-21): "...and then to multiply it with the height extent of fine particles in the column, derived by the collocated lidar measurements."

Page 8, line 22: Why 40,000? Is that an arbitrary or standard number of particles that modelers use, or was there another reason for that choice?

REPLY: In Lagrangian dispersion models the number of tracer particles defines the accuracy of the simulation. 40000 particles are assumed adequate for the current experiment.

Page 9, line 12: what is Eta in "24 Eta vertical layers"?

REPLY: Eta refers to the ETA vertical coordinate system used in the ETA/NCEP dynamical core of DREAM model. We refer to the Eta/NCEP atmospheric model in beginning of this paragraph, line 8.

Page 9, line 13: Change 1/3 to 0.33

REPLY: We changed the text accordingly.
Page 10, lines 3-9: I think it might be good to include a shorter version of this in the abstract since currently you do not mention anything about the models or in situ measurements that you use to aid in the characterization study.

REPLY: We included the following in the abstract (pg. 2, lines 14-15): "The results are also compared with modelled dust and marine concentration profiles and surface in situ measurements."

Page 10, line 28: How much is "quite well" in % difference?

REPLY: We included the following in the text (pg. 10, lines 30-31 and pg. 11, lines 1-2): "Our results show that GARRLiC and LIRIC backscatter and extinction coefficient profiles at 355, 532 and 1064 nm agree quite well, with their differences being 10-20% with respect to GARRLiC values, well within the LIRIC uncertainties (Fig. 4a and b)."

Figure 4: Consider labeling each panel like 4a, 4b, etc so you don't have to refer to them as "first and second row in Fig 4".

REPLY: We changed the text accordingly.

Page 11, line 1: replace "cut above" by "restricted to"

REPLY: We changed the text accordingly.

Page 12, line 7-8: "Moreover, due to the low RH at the surface (16%) we do not (...)"

REPLY: We changed the text accordingly.

Page 12, line 10: What's excellent in % difference? Please quantify it.

REPLY: We changed the text as following (pg. 12, lines 15-16): "The concentration profiles from GARRLiC and LIRIC are in excellent agreement at heights >1 km, with differences to be less than 10% (Fig. 6a)."

Figure 6: Instead of having Fig 6b with two panels, just relabel each plot in Figure 6 as 6a, 6b, 6c. Makes it easier to reference.
REPLY: We changed the text accordingly.

Page 13, line 1: replace "indicatory" by "indicative" or "qualitative"

REPLY: We changed the text accordingly.

Page 14, line 7: missing LR unit (sr).

REPLY: We changed the text accordingly.

Page 19, line 3-17: Please quantify the agreement levels that you mention in this paragraph. How many %?

REPLY: We changed the text as following (pg. 19, lines 7-16): "For the first case GAR-RLiC achieves a successful retrieval of the dust vertical distribution and microphysical characterization that agrees well with AERONET and climatological values for dust, within the respective uncertainties. Both LIRIC and GARRLiC concentration profiles are found to be consistent with the BSC DREAM8b dust vertical structure, showing though up to 100% larger values than the surface in situ PM10 measurements. For the second case consisting of mainly marine particles, both algorithms provide satisfactory concentration retrievals, well within the time variability of the surface in situ PM10 measurements. The GARRLiC microphysical property retrieval is mostly not successful for the marine particles, with e.g.  $\sim$ 10% more fine particle volume than the AERONET product and the surface in situ measurements."

Page 19, line 8: replace "from" by "than"

REPLY: We changed the text accordingly.

Page 19, line 24: replace "near-to-surface" by "near-surface".

REPLY: We changed the text accordingly.

---

## Author Comment (AC3) · 21 Aug 2017

Both active and passive remote sensing instruments have their own advantages and disadvantages in terms of the vertically-resolved retrievals of the aerosol microphysical parameters. It is clear that the combination of active/passive technologies is absolutely necessary in order to improve the quality of retrievals. Alexandra Tsekeri and coauthors are moving in that direction. Finokalia lidar station is famous to have a unique geographical location allowing observation of the different combinations of the aerosol particles from Central Europe and Africa/Sahara. It is nice to get a reminder that Finokalia station is operational and actively delivering the valuable scientific data related

to the dust and marine aerosols. The paper in overall is very well written. It is pleasure to read the text. The wording and grammar are close to be perfect. I recommend to accept this manuscript for the publication after a few minor corrections.

REPLY: We thank the reviewer for his/her kind words!

General comments:

1. After reading the paper I got an impression that there is some kind of luck that GARRLiC and LIRIC algorithms work and result in a reasonable aerosol microphysical retrievals. Here is the list of introduced assumptions to support this impression:

A) "The volume concentration below the lowest height of the lidar signals is considered to be constant" (page 4, line 9) It is very difficult to imagine the constant volume concentration in the first few hundreds meters above the ground. Please keep in mind that the comparisons with in situ are performed using the near-surface data.

B) The Raman signals from the lidar are such a useful piece of information about aerosols, but not used in the retrievals at all (see Fig. 1). There is only a plan to use the extinction optical coefficients in future.

C) "GARRLiC ... is able to retrieve only one refractive index for each mode." (page 16, line 14) "considering refractive index to be constant along the atmospheric column" (page 4, line 6) The assumption of only one refractive index per profile is very damaging for the whole idea of vertically-resolved microphysical retrievals. The aerosols of different nature (thus, different refractive indexes) are often reside in a different layers of the same vertical profile. Please consider in your future studies to eliminate this assumption.

D) Some additional difficulties on the top of that: - "In addition, as seen in Fig. 7a, most of the aerosol load is located below 1 km, where the lidar incomplete overlap region is located, which challenges even more the combined lidar/sun-photometer retrieval." (page 14, line 1) As a result, there is a nice agreement between GARRLiC and LIRIC

in terms of volume concentration (see Fig. 10a), very strong disagreement between GARRLiC and LIRIC in terms of ambient PM10 (up to 3-4 times or so, see Fig. 10b.right above 1 km), and than finally fair agreement between GARRLiC, LIRIC, and in situ in terms of dry PM10 (see Fig. 10b.right below 1 km). All these sudden turns are really thrilling! Please provide some explanation in the text.

REPLY: GARRLiC and LIRIC have their limitations in successfully characterizing the particle microphysical property profiles, as the reviewer points out. We have clearly stated this in the paper, e.g. in pg. 4, lines 26-30: "Both algorithms work well for individual aerosol components or mixtures of (mainly) fine (e.g. pollution) and (mainly) coarse (e.g. dust) particles, but they should not be able to fully characterize the mixture components in case of more than one fine or coarse mode in the mixture, as in smoke/pollution or dust/marine mixture cases.". Going through the reviewer's individual points:

A) We changed the text (pg. 4, lines 12-13): "The concentrations are considered constant below the lowest height of the lidar signals, which may introduce errors in the retrieved profiles (e.g. Tsekeri et al., 2013)."

B) The Raman signals are very useful for the retrieval, but they are not available for the day-time retrievals considered here, as of yet.

C) GARRLiC retrieves one refractive index for each mode: in case of two modes in the profile, the total refractive index may change along the column. We changed the text so as to highlight this point (pg. 4, lines 9-11): "Although in GARRLiC the microphysical properties are considered to be constant along the column for each mode, the total values change along the column in case of two modes with different properties." Moreover, we added a comment for the change in profile properties due to the particle hygroscopic growth, following the advice of reviewer #1 (pg. 4, lines 18-22): "In case of multi-mode aerosol mixtures and/or change of microphysical properties with height due to particle hygroscopic growth (e.g. Tsekeri et al., 2017) an inherent deficiency

of both algorithms is the number of aerosol modes retrieved, with LIRIC considering three modes (fine particles, coarse spherical and coarse non-spherical particles) and GARRLiC considering two modes (fine and coarse particles)."

D) We do not agree with the reviewer that there is nice agreement between GARRLiC and LIRIC in terms of volume concentration in Fig. 10a, for the retrieval below 1 km. More specifically, GARRLiC retrieves ∼100% more coarse particle volume concentration than LIRIC below 1 km. Concerning the disagreement in the ambient PM10: The ambient PM10 profile in Fig. 10c is the sum of fine and part of the coarse particle volume concentration, multiplied by the corresponding densities. The small and large differences seen below and above 1 km, respectively, are due to the differences in particle densities, with the fine particles to have a density of 1.8 g cm^(-3), which is 40% larger than the coarse particle density of 1.25 g cm^(-3).

2. GARRLiC and LIRIC algorithms are based on the usage of pre-calculated AERONET products. Without going too much into a details, they sound almost like a twins or, at least, share some part of the software code. For the final users at the lidar stations it is not convenient to have several twins-like algorithms in a package. It is confusing to have similar results for the one group of microphysical parameters and different results for the other group of parameters. Is there a plan to come up with the best way on how to combine the lidar/sun-photometer data that will merge/replace GARRLiC and LIRIC in a single algorithm? It is highly desirable if authors will share their vision regarding this issue in a paragraph of text.

REPLY: We are not aware of any future plans for combination/merging of GARRLiC and LIRIC algorithms. The algorithms are not as similar as the reviewer states here, as proven from the results shown for the three cases in the paper. For example, GARRLiC does not use any pre-calculated AERONET products. To avoid confusion, we highlighted this in the text by adding the following description for GARRLiC (pg. 4, lines 5-7): "The algorithm does not use the AERONET products, but it instead calculates the size distribution, spherical particle fraction and spectral complex refractive index,

separately for fine and coarse particles."

Specific comments:

1. Page 4. Line 4: "The algorithm calculates the size distribution, spherical particle fraction and spectral complex refractive index, separately for fine and coarse particles, considering them constant along the atmospheric column, and the volume concentration profiles of fine and coarse particles." This sentence is quite unclear and ambiguous. Please consider to split it into two simpler sentences.

REPLY: We did it. See comment above.

2. Page 11. Line 11 "m." instead of "m:"

REPLY: We changed the text accordingly.

---

## Referee Comment (RC4) · Anonymous Referee #3 · 30 Aug 2017

Authors addressed all the concerns that I raised in my review for their paper. I recommend publishing the paper of Tsekeri et al in its revised version. Good luck with your future studies of the mineral dust related particles at the dust hot spot of Europe!